# Mambular: A Sequential Model for Tabular Deep Learning

## Abstract

The analysis of tabular data has traditionally been dominated by gradient-boosted decision trees (GBDTs), known for their proficiency with mixed categorical and numerical features. However, recent deep learning innovations are challenging this dominance. We introduce Mambular, an adaptation of the Mamba architecture optimized for tabular data. We extensively benchmark Mambular against state-of-the-art models, including neural networks and tree-based methods, and demonstrate its competitive performance across diverse datasets. Additionally, we explore various adaptations of Mambular to understand its effectiveness for tabular data. We investigate different pooling strategies, feature interaction mechanisms, and bi-directional processing. Our analysis shows that interpreting features as a sequence and passing them through Mamba layers results in surprisingly performant models. The results highlight Mambular's potential as a versatile and powerful architecture for tabular data analysis, expanding the scope of deep learning applications in this domain. The source code is available at https://anonymous.4open.science/r/mamba-tabular-485F/.

## 1 Introduction

Gradient-boosted decision trees (GBDTs) have long been the dominant approach for analyzing tabular data, due to their ability to handle the typical mix of categorical and numerical features found in such datasets (Grinsztajn et al., 2022). In contrast, deep learning models have historically faced challenges with tabular data, often struggling to outperform GBDTs. The complexity and diversity of tabular data, including issues like missing values, varied feature types, and the need for extensive preprocessing, have made it difficult for deep learning to match the performance of GBDTs (Borisov et al., 2022). However, recent advancements in deep learning are gradually challenging this paradigm by introducing innovative architectures that leverage advanced mechanisms to capture complex feature dependencies, promising significant improvements (Popov et al., 2019; Hollmann et al., 2022; Gorishniy et al., 2021).

One of the most effective advancements in tabular deep learning is the application of attention mechanisms in models like TabTransformer (Huang et al., 2020), FT-Transformer (Gorishniy et al., 2021) and many more (Wang and Sun, 2022; Thielmann et al., 2024b; Arik and Pfister, 2021). These models leverage the attention mechanism to capture dependencies between features, offering a significant improvement over traditional approaches. FT-Transformers, in particular, have demonstrated robust performance across various tabular datasets, often surpassing the accuracy of GBDTs (McElfresh et al., 2024). Additionally, more traditional models like Multi-Layer Perceptrons (MLPs) and ResNets have demonstrated improvements when well-designed and when the data undergoes thorough preprocessing (Gorishniy et al., 2021; 2022). These models have benefited especially from innovations in advanced preprocessing methods that make them more competitive.

More recently, the Mamba architecture (Gu and Dao, 2023) has shown promising results in textual problems. Tasks previously dominated by Transformer architectures, such as DNA modeling and language modeling, have seen improvements with the application of Mamba models (Gu and Dao, 2023; Schiff et al., 2024; Zhao et al., 2024). Several adaptations have demonstrated its versatility, such as Vision Mamba for image classification (Xu et al., 2024), video analysis (Yang et al., 2024; Yue and Li, 2024) and point cloud analysis (Zhang et al., 2024; Liu et al., 2024). Furthermore, the

architecture has been adapted for time series problems, with notable successes reported by Patro and Agneeswaran (2024), Wang et al. (2024) and Ahamed and Cheng (2024b). Mamba has also been integrated into graph learning (Behrouz and Hashemi, 2024) and imitation learning (Correia and Alexandre, 2024). Further advancements have improved the language model, for example, by incorporating attention (Lieber et al., 2024), Mixture of Experts (Pióro et al., 2024) or bi-directional sequence processing (Liang et al., 2024).

These advancements underscore Mamba's broad applicability, making it a powerful and flexible architecture for diverse tasks and data types. Similarly to the transformer architecture, the question arises whether the Mamba architecture can also be leveraged for tabular problems, and this study is focused on addressing this question.

The contributions of the paper can be summarized as follows:

    I. We introduce Mambular, a tabular adaptation of Mamba, demonstrating the potential of sequential models in addressing tabular problems.

    II. We conduct extensive benchmarking of Mambular against several competitive neural and tree-based methods, illustrating that a standard Mambular model performs on par with or better than tree-based models across a wide range of datasets.

    III. We examine the impact of bi-directional processing and feature interaction layers on Mambular's performance, and compare several pooling methods.

    IV. Finally, we carry out an in-depth analysis of Mambular's sequential nature, investigating the implications of feature orderings in a sequential tabular model.

## 2 METHODOLOGY

For a tabular problem, let $\mathcal{D} = \{(\boldsymbol{x}^{(i)}, y^{(i)})\}_{i=1}^{n}$ be the training dataset of size $n$ and let $y$ denote the target variable that can be arbitrarily distributed. Each input $\boldsymbol{x} = (x_1, x_2, \ldots, x_J)$ contains $J$ features (variables). Categorical and numerical features are distinguished such that $\boldsymbol{x} \equiv (\boldsymbol{x}_{cat}, \boldsymbol{x}_{num})$, with the complete feature vector denoted as $\boldsymbol{x}$. Further, let $x_{j(cat)}^{(i)}$ denote the $j$-th categorical feature of the $i$-th observation, and hence $x_{j(num)}^{(i)}$ denote the $j$-th numerical feature of the $i$-th observation.

Following standard tabular transformer architectures, the categorical features are first encoded and embedded. In contrast to classical language models, each categorical feature has its own, distinct vocabulary to avoid problems with binary or integer encoded variables. Including <UNK> tokens additionally allows to easily deal with unknown or missing categorical values during training or inference.

Numerical features are mapped to the embedding space via a simple linear layer. However, since a single linear layer does not add information beyond a linear transformation, Periodic Linear Encodings, as introduced by Gorishniy et al. (2022) are used for all numerical features. Thus, each numerical feature is encoded before being passed through the linear layer for rescaling. Simple decision trees are used for detecting the bin boundaries, $b_t$, and depending on the task, either classification or regression is employed for the target-dependent encoding function $h_j(\boldsymbol{x}_{j(num)}, y)$. Let $b_t$ denote the decision boundaries from the decision trees. The encoding function is given in Eq. 1.

**PLE**

$$z_{j(\text{num})}^{t} = \begin{cases} 0 & \text{if } x < b_{t-1}, \\ 1 & \text{if } x \geq b_t, \\ \frac{x - b_{t-1}}{b_{t-2} - b_{t-1}} & \text{else.} \end{cases} \tag{1}$$

The feature encoding and embedding generation is demonstrated in Figure 1. The created embeddings, following classical statistical literature (Hastie et al., 2009; Kneib et al., 2023) are denoted as $\mathbf{Z}$ and not $\mathbf{X}$ to clarify the difference between the embeddings and the raw features.

Subsequently, the embeddings are passed jointly through a stack of Mamba layers. These include one-dimensional convolutional layers to account for invariance of feature ordering in the pseudo-sequence as well as a state-space (SSM) model (Gu et al., 2021; Hamilton, 1994). The feature matrix

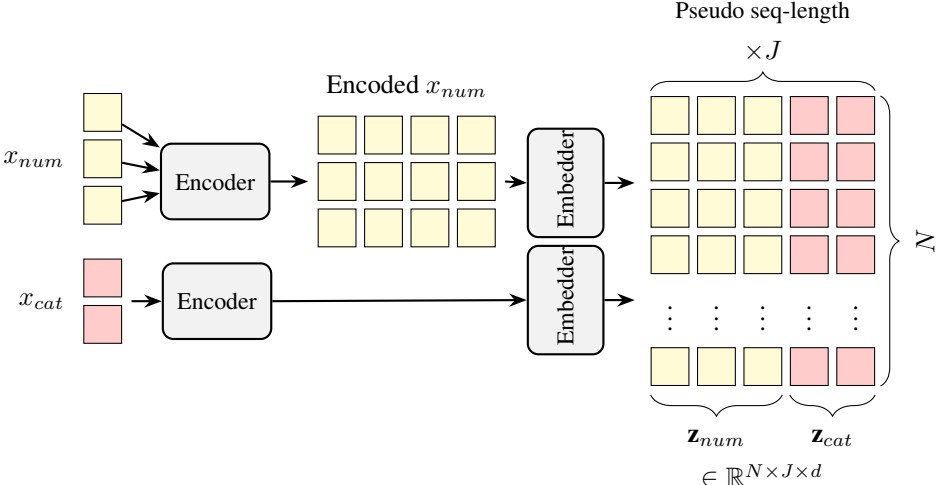

Figure 1: Generation of the input matrix that are fed through the Mamba blocks. The categorical features are tokenized and embedded similar to classical embeddings for language models. The numerical features are encoded and embedded via a simple linear layer. The final input matrix of the Mamba blocks are the concatenated embeddings $\mathbf{z} \in \mathbb{R}^{N \times J \times d}$ with embedding dimension $d$.

before being passed through the SSM model has a shape of (BATCH SIZE) $\times$ J $\times$ (EMBEDDING DIMENSION), later referenced as $N \times J \times d$. Importantly, the sequence length in a tabular context refers to the number of variables, and hence the second dimension, $J$, corresponds to the number of features rather than to the length of, e.g., a document.

The convolution operation along the sequence length $J$ and with Kernel $K$ is expressed as:

$$\mathbf{Z}_{\text{conv}}^{(n,d)}(j) = \sum_{m=0}^{K-1} \mathbf{Z}^{(n,d)}[j+m] \cdot \mathbf{k}^{(d)}(m),$$
$$\forall n \in \{1, \ldots, N\}, \forall d \in \{1, \ldots, d\}, \forall j \in \{1, \ldots, J - K + 1\},$$

where $\mathbf{Z}_{\text{conv}}^{(n,d)}(j)$ is the $j$-th element of the convolved sequence for batch $n$ and feature channel $d$. $\mathbf{Z}^{(n,d)}[j+m]$ is the $[j+m]$-th element of the input sequence $\mathbf{Z}$ for batch $n$ and feature channel $d$, and $K$ describes the kernel size. Summing over the elements of the kernel, indexed by $m$, accounts for the variable position in the pseudo-sequence. Thus, setting the kernel size equivalent to the number of variables would make the sequence invariant positional permutations. The resulting output tensor retains the same shape as the input, since padding is set to the kernel size -1.

After the convolution, given the matrices:

$$\mathbf{A} \in \mathbb{R}^{1 \times 1 \times d \times \delta}, \quad \mathbf{B} \in \mathbb{R}^{N \times J \times 1 \times \delta}, \quad \Delta \in \mathbb{R}^{N \times J \times d \times 1}, \bar{\mathbf{z}} \in \mathbb{R}^{N \times J \times d \times 1},$$

where $\delta$ denotes a inner dimension, similar to the feed forward dimension in Transformer architectures and $\bar{\mathbf{z}}$ has the same entries as $\mathbf{z}$, but one additional axis, the formula for updating the hidden state $\mathbf{h}_j \in \mathbb{R}^{N \times d \times \delta}$ is:

$$\mathbf{h}_j = \exp\left(\Delta \odot_3 \mathbf{A}\right)_{:,j,:,:} \odot_{1,2,3} \mathbf{h}_{j-1} + \left((\Delta \odot_{1,2} \mathbf{B}) \odot_{1,2,3} \bar{\mathbf{z}}\right)_{:,j,:,:}. \tag{2}$$

The symbol $\odot_d$ denotes an outer product where the multiplication is done for the $d$-th axis and parallelized wherever a singleton axis length meets an axis of length one[1]. The exponential function is applied element-wise. The state transition matrix $\mathbf{A}$ governs the transformation of the hidden state from the previous time step to the current one, capturing how the hidden states evolve independently of the input features. The input-feature matrix $\mathbf{B}$ maps the input features to the hidden state space, determining how each feature influences the hidden state at each step. The gating matrix $\mathbf{\Delta}$ acts as

---

[1]This corresponds to using the ordinary multiplication operator "*" in PyTorch and relying on the default broadcasting

a gating mechanism, modulating the contributions of the state transition and input-feature matrices, and allowing the model to control the extent to which the previous state and the current input affect the current hidden state.

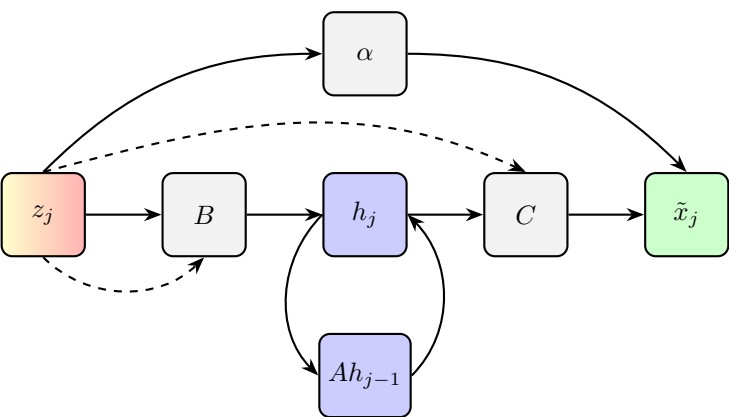

Figure 2: SSM updating step with recursive update of $h$: The hidden state is iteratively updated by going through the sequence (features) similar to a recurrent neural network. The final representation is generated as described in Equations 3-4.

In contrast to FT-Transformer (Gorishniy et al., 2021) and TabTransformer (Huang et al., 2020) Mambular truly iterates through all variables as if they are a sequence; hence, feature interactions are detected sequentially. The effect of feature position in a sequence, and the impact of the convolution kernel size is analyzed with respect to performance in section 4. Furthermore, it should be noted that in contrast to TabPFN (Hollmann et al., 2022), Mambular does not transpose dimensions and iterates over observations. Hence, training on large datasets is possible and it can scale well to any training data size, just as Mamba (Gu and Dao, 2023) does.

After stacking and further processing, the final representation, $\tilde{\mathbf{x}} \in \mathbb{R}^{N \times J \times d}$ is retrieved. In truly sequential data, these are the contextualized embeddings of the input tokens, for tabular problems $\tilde{\mathbf{x}}$ represents a contextualized, or feature interaction accounting variable representation, in the embedding space. The hidden states are stacked along the sequence dimension to form:

$$\mathbf{H} = [\mathbf{h}_0, \mathbf{h}_1, \ldots, \mathbf{h}_{T-1}] \in \mathbb{R}^{N \times J \times d \times \delta}.$$

The final output representation $\tilde{\mathbf{x}}$ is then computed by performing matrix multiplication of the stacked hidden states with matrix $\mathbf{C} \in \mathbb{R}^{N \times J \times 1 \times \delta}$ where the multiplication and summation is done over the last axis, and adding the vector $\alpha \in \mathbb{R}^{1 \times 1 \times d}$ scaled by the input $\mathbf{z}$:

$$\tilde{\mathbf{x}} = (\mathbf{H} \cdot_4 \mathbf{C}) + (\alpha \odot_3 \mathbf{z}). \tag{3}$$

More explicitly, this can be written as:

$$\tilde{x}_{i,j,k} = \sum_{\delta} \mathbf{H}_{i,j,k,\delta} \mathbf{C}_{i,j,1,\delta} + \alpha_{1,1,k} \mathbf{z}_{i,j,k}.$$

where $\mathbf{C}$ and $\alpha$ are learnable parameters. For final processing, $\tilde{\mathbf{x}}$ is element-wise multiplied with $\mathbf{z}'$, and the result is passed through a final linear layer:

$$\tilde{\mathbf{x}}_{\text{final}} = (\tilde{\mathbf{x}} \odot_{1,2,3} \mathbf{z}') \mathbf{W}_{\text{final}} + \mathbf{b}_{\text{final}}. \tag{4}$$

Pooling is an important step before passing $\tilde{\mathbf{x}}_{\text{final}}$ to the final task specific model head. Average pooling is the method that mambular is taking advantage of for this phase. Other pooling methods has been evaluated in the section 4.

The model is trained end-to-end by minimizing the task-specific loss, e.g., mean squared error for regression or categorical cross entropy for classification tasks. An overview of a forward pass in the model is given in Figure 2.

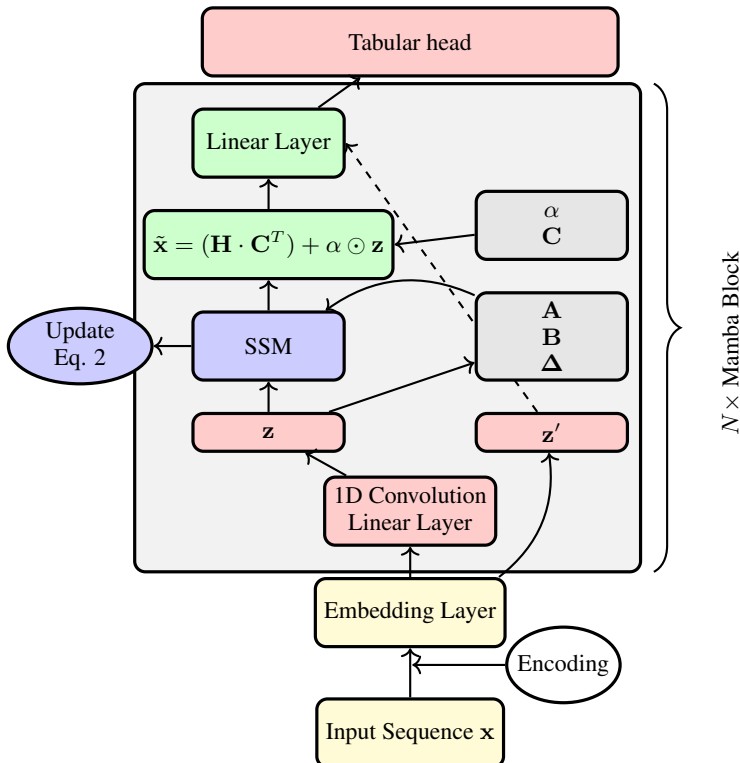

Figure 3: The forward pass of a single sequence in the model. After embedding the inputs, the embeddings are passed to several Mamba blocks. The tabular head is a single task specific output layer. Before being passed to the Linear Layer, the contextualized embeddings are pooled via average pooling. For bidirectional processing a second block with a flipped sequence is used and the learnable matrices are not shared between the directions.

## 3 EXPERIMENTS

Mambular is evaluated against a range of top-performing models (McElfresh et al., 2024) across multiple datasets (Supplementary Table 8). These models include FT-Transformer (Gorishniy et al., 2021), TabTransformer (Huang et al., 2020), XGBoost (Grinsztajn et al., 2022; McElfresh et al., 2024), LightGBM (Ke et al., 2017), a Random Forest, a baseline Multi-Layer Perceptron, and a ResNet. TabPFN (Hollmann et al., 2022) is excluded due to its unsuitability for larger datasets.

A 5-fold cross-validation is conducted for all datasets, with average results and standard deviations reported. PLE encodings (Eq. 1) with a maximum number of bins equal to the model dimension are used for all neural models (128 for most models, including MLP and ResNet). All categorical features are integer-encoded. For regression tasks, targets are normalized. Mean Squared Error (MSE) and Area Under the Curve (AUC) metrics are reported for regression and classification tasks respectively. TabTransformer, FT-Transformer, and Mambular employ identical architectures for embeddings and task-specific heads, which includes a single output layer without activation function or dropout. The [CLS] token embedding is utilized for final prediction in the FT-Transformer as it has been shown to enhance performance (Thielmann et al., 2024b; Gorishniy et al., 2021).

All neural models share several parameters: a starting learning rate of 1e-04, weight decay of 1e-06, an early stopping patience of 15 epochs with respect to the validation loss, a maximum of 200 epochs for training, and learning rate decay with a factor of 0.1 with a patience of 10 epochs with respect to the validation loss. A universal batch size of 128 is used, and the best model with respect to the validation loss is returned for testing. TabTransformer, FT-Transformer, and Mambular use the same embedding functions. For the benchmarks, a basic Mambular architecture is employed, using average pooling, no feature interaction layer, and no bi-directional processing. The columns/sequence are always sorted with numerical features first, followed by categorical features. Within these two

groups, the features are sorted as they were originally provided in the dataset from the UCI Machine Learning Repository. A small kernel size of 4 in the convolutional layer is used based on the default Mamba architecture. The impact of variable positioning (with respect to the kernel size) on sequential processing is analyzed in section 4. Details on the used datasets and preprocessing can be found in Appendix A. Details on the model architectures and hyperparameters can be found in Appendix E.

**Comparison to XGBoost**   When benchmarked against XGBoost using default hyperparameter settings, Mambular demonstrates comparable, if not slightly superior performance. It significantly outperforms XGBoost on 4 out of 12 datasets at the 10% significance level, while XGBoost surpasses Mambular on 2 datasets at the same significance level. The $p$-values from simple t-tests over the folds are reported for each dataset with testing methodology based on Gorishniy et al. (2021).

After adjusting for multiple testing via Benjamini-Hochberg (Ferreira and Zwinderman, 2006; Benjamini and Hochberg, 1995) the Abalone results - only significant at the 10% level with standard testing - are not significant anymore. All other results remain unchanged[2].

Table 1: Comparison between Mambular and XGBoost. The left tables shows regression results with average MSE values over 5 folds. The right side shows (binary) classification results with average AUC values. Significantly better values at the 5% significance level are in green and marked bold. Significantly better values at the 10% significance level are underscored. Dataset details can be found in appendix A. ↑ depicts higher is better and vice-versa.

| Models | DI ↓ | AB ↓ | CA ↓ | WI ↓ | PA ↓ | HS ↓ | CP ↓ | BA ↑ | AD ↑ | CH ↑ | FI ↑ | MA ↑ |
|---|---|---|---|---|---|---|---|---|---|---|---|---|
| Mambular | **0.018** | 0.452 | 0.167 | 0.628 | 0.035 | 0.132 | 0.025 | 0.927 | 0.928 | **0.861** | **0.796** | 0.917 |
| XGB | 0.019 | 0.506 | 0.171 | **0.528** | 0.036 | 0.119 | 0.024 | 0.928 | 0.929 | 0.845 | 0.774 | **0.922** |
| $p$-value | 0.0079 | 0.0870 | 0.4865 | 1.3e-07 | 0.6287 | 0.3991 | 0.1999 | 0.7883 | 0.7930 | 0.0192 | 0.0120 | 0.010 |

**Overall Performance**   Table 2 provides a comprehensive ranking of all evaluated methods and their performance in both regression and classification tasks. The results align with existing literature, highlighting the strong performance of the FT-Transformer architecture (Gorishniy et al., 2021), LightGBM, CatBoost and XGBoost (McElfresh et al., 2024).CatBoost emerges as the overall best-performing model across all tasks. Among the evaluated models, Mambular stands out as the top-performing neural model on average across all datasets. Additional benchmark results, including additional datasets can be found in Appendix F.

Table 2: Combined Ranking of Models for Regression and Classification Tasks. The best model is marked in bold and second best in italic. CatBoost is the overall best performing model, followed by Mambular. Mambular is the best model among all deep learning architectures.

| Models | | Regression Rank | Classification Rank | Overall Rank |
|---|---|---|---|---|
| Trees | XGBoost | $4.57 \pm 2.57$ | $4.6 \pm 3.29$ | $4.58 \pm 2.75$ |
| | RF | $4.57 \pm 2.37$ | $6.6 \pm 2.07$ | $5.42 \pm 2.39$ |
| | LightGBM | $4.29 \pm 1.60$ | $3.2 \pm 2.95$ | $3.83 \pm 2.21$ |
| | CatBoost | $3.71 \pm 2.29$ | $\mathbf{2.2} \pm 1.10$ | $\mathbf{3.08} \pm 1.98$ |
| Neural | FT-Transformer | $\mathbf{3.14} \pm 1.86$ | $4.6 \pm 1.52$ | $3.75 \pm 1.82$ |
| | MLP | $9.00 \pm 0.82$ | $7.8 \pm 2.95$ | $8.50 \pm 1.98$ |
| | TabTransformer | $9.20 \pm 0.84$ | $8.0 \pm 1.41$ | $8.67 \pm 1.22$ |
| | ResNet | $7.14 \pm 2.04$ | $7.0 \pm 2.55$ | $7.08 \pm 2.15$ |
| | NODE | $5.29 \pm 2.63$ | $7.2 \pm 1.64$ | $6.08 \pm 2.39$ |
| | Mambular | $3.71 \pm 2.63$ | $3.0 \pm 1.22$ | $3.42 \pm 2.11$ |

Detailed results for all datasets and tasks can be found in Table 3 and 4, with additional results on further models provided in Appendix F. Notably, all neural models underperform on the Wine dataset, while XGBoost lags behind all neural models on the Abalone and FICO datasets. Our findings

---

[2]Due to the small sample sizes, Benjamini-Hochberg is preferred to the conservative Bonferroni adjustments (Nakagawa, 2004).

also indicate that both FT-Transformer and Mambular excel on datasets with very few categorical features (e.g., FICO, California Housing, Abalone, CPU), despite their designs being optimized for discrete data inputs.

Table 3: Benchmarking results for the regression tasks. Average mean squared error values over 5 folds and the corresponding standard deviations are reported. Smaller values are better. The best performing model is marked in bold.

| Models | DI ↓ | AB ↓ | CA ↓ | WI ↓ | PA ↓ | HS ↓ | CP ↓ |
|---|---|---|---|---|---|---|---|
| XGBoost | $0.019 \pm 0.000$ | $0.506 \pm 0.044$ | $0.171 \pm 0.007$ | $0.528 \pm 0.008$ | $0.036 \pm 0.004$ | $0.119 \pm 0.024$ | $0.024 \pm 0.004$ |
| RF | $0.019 \pm 0.001$ | $0.461 \pm 0.052$ | $0.183 \pm 0.008$ | $\mathbf{0.485} \pm 0.007$ | $0.028 \pm 0.006$ | $0.121 \pm 0.018$ | $0.025 \pm 0.002$ |
| LightGBM | $0.019 \pm 0.001$ | $0.459 \pm 0.047$ | $0.171 \pm 0.007$ | $0.542 \pm 0.013$ | $0.039 \pm 0.007$ | $0.112 \pm 0.018$ | $0.023 \pm 0.003$ |
| CatBoost | $0.019 \pm 0.000$ | $0.457 \pm 0.007$ | $0.169 \pm 0.006$ | $0.583 \pm 0.006$ | $0.045 \pm 0.005$ | $\mathbf{0.106} \pm 0.015$ | $\mathbf{0.022} \pm 0.001$ |
| FT-Transformer | $0.018 \pm 0.001$ | $0.458 \pm 0.055$ | $0.169 \pm 0.006$ | $0.615 \pm 0.012$ | $\mathbf{0.024} \pm 0.005$ | $0.111 \pm 0.014$ | $0.024 \pm 0.001$ |
| MLP | $0.066 \pm 0.003$ | $0.462 \pm 0.051$ | $0.198 \pm 0.011$ | $0.654 \pm 0.013$ | $0.764 \pm 0.023$ | $0.147 \pm 0.017$ | $0.031 \pm 0.001$ |
| TabTransformer | $0.065 \pm 0.002$ | $0.472 \pm 0.057$ | $0.247 \pm 0.013$ | - | $0.135 \pm 0.001$ | $0.160 \pm 0.028$ | - |
| ResNet | $0.039 \pm 0.018$ | $0.455 \pm 0.045$ | $0.178 \pm 0.006$ | $0.639 \pm 0.013$ | $0.606 \pm 0.031$ | $0.141 \pm 0.017$ | $0.030 \pm 0.002$ |
| NODE | $0.019 \pm 0.000$ | $\mathbf{0.431} \pm 0.052$ | $0.207 \pm 0.001$ | $0.613 \pm 0.006$ | $0.045 \pm 0.007$ | $0.124 \pm 0.015$ | $0.026 \pm 0.001$ |
| Mambular | $\mathbf{0.018} \pm 0.000$ | $0.452 \pm 0.043$ | $\mathbf{0.167} \pm 0.011$ | $0.628 \pm 0.010$ | $0.035 \pm 0.005$ | $0.132 \pm 0.020$ | $0.025 \pm 0.002$ |

Table 4: Benchmarking results for the classification tasks. Average AUC values over 5 folds and the corresponding standard deviations are reported. Larger values are better.

| Models | BA ↑ | AD ↑ | CH ↑ | FI ↑ | MA ↑ |
|---|---|---|---|---|---|
| XGBoost | $0.928 \pm 0.004$ | $\mathbf{0.929} \pm 0.002$ | $0.845 \pm 0.008$ | $0.774 \pm 0.009$ | $0.922 \pm 0.002$ |
| RF | $0.923 \pm 0.006$ | $0.896 \pm 0.002$ | $0.851 \pm 0.008$ | $0.789 \pm 0.012$ | $0.917 \pm 0.004$ |
| LightGBM | $\mathbf{0.932} \pm 0.004$ | $0.929 \pm 0.001$ | $0.861 \pm 0.008$ | $0.788 \pm 0.010$ | $\mathbf{0.927} \pm 0.001$ |
| CatBoost | $0.932 \pm 0.008$ | $0.927 \pm 0.002$ | $\mathbf{0.867} \pm 0.006$ | $0.796 \pm 0.010$ | $0.926 \pm 0.005$ |
| FT-Transformer | $0.926 \pm 0.003$ | $0.926 \pm 0.002$ | $0.863 \pm 0.007$ | $0.792 \pm 0.011$ | $0.916 \pm 0.003$ |
| MLP | $0.895 \pm 0.004$ | $0.914 \pm 0.002$ | $0.840 \pm 0.005$ | $0.793 \pm 0.011$ | $0.886 \pm 0.003$ |
| TabTransformer | $0.921 \pm 0.004$ | $0.912 \pm 0.002$ | $0.835 \pm 0.007$ | - | $0.910 \pm 0.002$ |
| ResNet | $0.896 \pm 0.006$ | $0.917 \pm 0.002$ | $0.841 \pm 0.006$ | $0.793 \pm 0.013$ | $0.889 \pm 0.003$ |
| NODE | $0.914 \pm 0.008$ | $0.904 \pm 0.002$ | $0.851 \pm 0.006$ | $0.790 \pm 0.010$ | $0.904 \pm 0.005$ |
| Mambular | $0.927 \pm 0.006$ | $0.928 \pm 0.002$ | $0.861 \pm 0.008$ | $\mathbf{0.796} \pm 0.013$ | $0.917 \pm 0.003$ |

**Distributional Regression**   To further validate Mambular's suitability for tabular problems, we conducted a small task on distributional regression (Kneib et al., 2023). Mambular for Location Scale and Shape (MambularLSS) outperforms XGBoostLSS (März, 2019) in terms of Continuous Ranked Probability Score (CRPS) (Gneiting and Raftery, 2007) when minimizing the negative log-likelihood while maintaining a small MSE. A detailed analysis can be found in Appendix C.

## 4   ABLATION

**Model Architecture**   This section explores the impact of various elements of Mambular's architecture, including (i) different pooling techniques, (ii) interaction layers, and (iii) bidirectional processing (Table 5). Transformer networks for natural language processing often use [CLS] token embeddings for pooling (Gorishniy et al., 2021), a technique that has also proven beneficial in tabular problems (Thielmann et al., 2024b). Therefore, this technique is evaluated here. For pooling techniques, we compared Sum-pooling, Max-pooling, Last token pooling – where only the last token in the sequence is passed to the task-specific model head –, and [CLS] pooling[3] against standard Average-pooling.

Given the significance of feature interactions in tabular problems, we also assessed the effectiveness of a learnable interaction layer between the features. This layer learns an interaction matrix $\mathbf{W} \in \mathbb{R}^{J \times J}$, such that interactions $= \mathbf{z}\mathbf{W}$, where $\mathbf{z}$ is the input feature matrix, before being passed through the SSM. This evaluation was only implemented for the standard Average pooling technique.

Interestingly, none of the configurations outperformed the basic architecture of average pooling, no interaction, and one-directional processing. Among the pooling strategies, last token pooling and

---

[3]Note that [CLS] token is appended to the end of each sequence in this implementation.

Table 5: Mean AUC and Mean MSE for various datasets and model configurations. We test different pooling methods, bi-directional processing and a learnable interaction layer. Significantly worse results compared to the default (average pooling, no interaction and no bi-directional processing) are marked red and bold at the 5% significance level and underscored and red at the 10% significance level. All results are achieved with 5-fold cross validation with identical seeds to the main results.

| Pooling | bi-directional | Interaction | BA ↑ | AD ↑ | AB ↓ | CA ↓ |
|---------|---------------|-------------|------|------|------|------|
| Last | × | × | **0.916** ± 0.004 | 0.927 ± 0.002 | 0.449 ± 0.043 | 0.181 ± 0.012 |
| Sum | × | × | 0.925 ± 0.005 | 0.928 ± 0.002 | 0.449 ± 0.048 | 0.171 ± 0.009 |
| Max | × | × | 0.928 ± 0.004 | 0.927 ± 0.002 | 0.455 ± 0.050 | 0.172 ± 0.008 |
| [CLS] | × | × | **0.914** ± 0.005 | 0.928 ± 0.002 | 0.478 ± 0.044 | **0.194** ± 0.018 |
| Avg | ✓ | × | 0.927 ± 0.004 | 0.928 ± 0.002 | 0.450 ± 0.045 | 0.170 ± 0.010 |
| Avg | × | ✓ | 0.928 ± 0.004 | 0.928 ± 0.002 | 0.453 ± 0.046 | 0.170 ± 0.007 |
| Avg | × | × | 0.927 ± 0.006 | 0.928 ± 0.002 | 0.452 ± 0.043 | 0.167 ± 0.011 |

[CLS] token pooling performed significantly worse on two out of the four tested datasets. For this ablation study, a 5-fold cross-validation was performed, with the same hyperparameters used across all models. In bi-directional processing, each direction has its own set of learnable parameters, meaning that bi-directional models have additional trainable parameters. All model configurations can be found in Appendix E.

**Sequence ordering** Unlike models that leverage attention layers, Mambular is a sequential model. However, tabular data is not inherently sequential – i.e., the order of features in tabular datasets should not matter. Therefore, we examined the significance of variables' position within the sequence and how their order impacts model performance. In textual data, shuffling the order of words/tokens significantly affects the outcome, and even swapping single words can lead to entirely different contextualized embeddings. Since these contextualized representations are pooled and fed directly to Mambular's task-specific head, this could also impact performance.

Evaluation experiments were conducted on four real-world datasets and simulated data (see Appendix B). As illustrated in Table 6, we initially confirmed the impact of the kernel size on tabular problems using Mamba's default kernel size of 4. The findings indicate that the order of sequences does not significantly influence model performance at the 5% level for the selected datasets, even with a relatively small kernel size. However, this is contingent on the data. Strong interaction effects between features that are positioned further apart than the kernel size in the pseudo-sequence can negatively impact model performance, as demonstrated by the results on the California housing dataset.

Table 6: Mean AUC and Mean MSE for different feature orderings in the sequence. Flipping. the sequence does not significantly affect the performance at the 5% or 10% significance level. Significantly different values at the 5% level from the default configuration (Num|Cat) are in bold and marked **red**.

| Model | BA ↑ | AD ↑ | AB ↓ | CA ↓ |
|-------|------|------|------|------|
| Num\|Cat | 0.927 ± 0.006 | 0.928 ± 0.002 | 0.452 ± 0.043 | 0.167 ± 0.011 |
| Cat\|Num | 0.925 ± 0.004 | 0.927 ± 0.002 | 0.454 ± 0.044 | 0.158 ± 0.007 |
| random shuffle | 0.923 ± 0.002 | 0.927 ± 0.002 | 0.457 ± 0.045 | 0.172 ± 0.070 |
| random shuffle | 0.921 ± 0.005 | 0.927 ± 0.002 | 0.459 ± 0.049 | 0.177 ± 0.010 |
| random shuffle | 0.924 ± 0.005 | 0.927 ± 0.002 | 0.453 ± 0.045 | **0.190** ± 0.010 |

The positions of the variables *Longitude* and *Latitude* appear to directly affect model performance (Table 7). Performance begins to decline significantly when *Longitude* and *Latitude* are outside the kernel window. This issue can be entirely resolved by increasing the kernel size to match the sequence length $J$. For a comprehensive analysis, refer to Appendix B.

Table 7: Analysis of results for CA Housing. Significantly worse results than the default ordering - numerical features: categorical features - and a kernel size of 4, are marked in red. Increasing the kernel size induces positional invariance for features within the sequence.

| Model | Kernel=4 ↓ | Kernel=J | Ordering |
|---|---|---|---|
| Num\|Cat | $0.167 \pm 0.011$ | - | [LO, LA, MA, TR, TB, Po, Hh, MI, OP] |
| Cat\|Num | $0.158 \pm 0.007$ | - | [OP, MI, Hh, Po, TB, TR, MA, LA, LO] |
| | $0.177 \pm 0.007$ | $0.160 \pm 0.007$ | [LO, MA, LA, TR, TB, Po, Hh, MI, OP] |
| | $0.175 \pm 0.008$ | $0.173 \pm 0.009$ | [LO, MA, TR, LA, TB, Po, Hh, MI, OP] |
| | $\mathbf{0.194} \pm 0.010$ | $0.169 \pm 0.008$ | [LO, MA, TR, TB, LA, Po, Hh, MI, OP] |
| | $\mathbf{0.196} \pm 0.011$ | $0.161 \pm 0.012$ | [LO, MA, TR, TB, Po, LA, Hh, MI, OP] |
| | $\mathbf{0.194} \pm 0.011$ | $0.173 \pm 0.009$ | [LO, MA, TR, TB, Po, Hh, LA, MI, OP] |
| | $\mathbf{0.195} \pm 0.010$ | $0.169 \pm 0.009$ | [LO, MA, TR, TB, Po, Hh, MI, LA, OP] |
| | $\mathbf{0.194} \pm 0.012$ | $0.172 \pm 0.011$ | [LO, MA, TR, TB, Po, Hh, MI, OP, LA] |

## 5 LIMITATIONS

The model we have presented has been tested across various datasets and compared against a range of models. However, we have not conducted hyperparameter tuning, as findings from Grinsztajn et al. (2022) and Gorishniy et al. (2021) suggest that most models perform adequately without tuning. These studies indicate that while hyperparameter tuning can enhance performance across all models simultaneously, it does not significantly alter the relative ranking of the models. This suggests that a model that performs best or worst with default configurations will likely retain its ranking even after extensive tuning. Furthermore, McElfresh et al. (2024) reported similar findings, strengthening the notion that hyperparameter tuning benefits most models equally without changing their comparative performance.

The absence of tuning does leave potential for enhancement across all models. However, the default configurations for the comparison models have been extensively tested in numerous studies. It is anticipated that if any model could gain more from hyperparameter tuning, it would be Mambular, due to the lack of extensive literature guiding its default settings. For the comparison models, we made our selections based on literature to ensure default parameters that are meaningful and high-performing. We managed to replicate average results from studies such as Gorishniy et al. (2021) and Grinsztajn et al. (2022). Moreover, key hyperparameters like learning rate, patience, and number of epochs are shared among all models for a more uniform approach. All hyperparameter configurations can be found in Appendix E.

## 6 CONCLUSION

We introduce Mambular, a novel architecture for tabular deep learning. Our work demonstrates the applicability of a genuinely sequential model to tabular problems, providing a unique viewpoint on the interpretation and management of tabular data by treating it as a sequential problem. Our findings indicate that a sequential model is effective for both regression and classification tasks across a variety of datasets. The performance of Mambular, along with its extension to MambularLSS, demonstrates its broad applicability to a wide range of tabular tasks.

While Mamba is still relatively new compared to architectures like the Transformer, its rapid adoption indicates substantial potential for further enhancement. Developments such as those proposed by Lieber et al. (2024) and Wang et al. (2024) could be particularly beneficial for tabular applications. Additionally, investigating the optimal feature ordering or integrating column-specific information through textual embeddings could further boost performance. Viewing tabular data as a sequence offers significant benefits for feature incremental learning. New features can be directly appended to the sequence, eliminating the need to retrain the entire model.

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

## A  DATASETS

All used datasets are taken from the UCI Machine Learning repository and publicly available. We drop out all missing values. For the regression tasks we standard normalize the targets. Otherwise, preprocessing is performed as described above. Note, that before PLE encoding we scale the numerical features to be within (-1, +1).

Table 8: The used datasets for benchmarking. All datasets are taken from the UCI Machine Learning repository. #num and #cat represent the number of numerical and categorical features respectively. The number of features thus determines for Mambular the "sequence length". The train, test and validation numbers represent the average number of samples in the respective split for the 5-fold cross validation. Ratio marks the percentage of the dominant class for the binary classification tasks.

| Name | Abbr. | #cat | #num | train | test | val | ratio |
|------|-------|------|------|-------|------|-----|-------|
| | | | | Regression Datasets | | | |
| Diamonds | DI | 4 | 7 | 34522 | 10788 | 8630 | - |
| Abalone | AB | 1 | 8 | 2673 | 835 | 668 | - |
| California Housing | CA | 1 | 9 | 13210 | 4128 | 3302 | - |
| Wine Quality | WI | 0 | 12 | 4158 | 1299 | 1039 | - |
| Parkinsons | PA | 2 | 20 | 3760 | 1175 | 940 | - |
| House Sales | HS | 8 | 19 | 13832 | 4322 | 3458 | - |
| CPU small | CPU | 0 | 13 | 5243 | 1638 | 1310 | - |
| | | | | Classification Datasets | | | |
| Bank | BA | 13 | 8 | 28935 | 9042 | 7233 | 88.3% |
| Adult | AD | 9 | 6 | 31259 | 9768 | 7814 | 76.1% |
| Churn | CH | 3 | 9 | 6400 | 2000 | 1600 | 79.6% |
| FICO | FI | 0 | 32 | 6694 | 2091 | 1673 | 53.3% |
| Marketing | MA | 15 | 8 | 27644 | 8638 | 6910 | 88.4% |

## B  SEQUENCE ORDERING

We test two different shuffling settings: **i)** shuffling the embeddings after they have passed through the embedding layer, **ii)** shuffling the sequence of variables before being passed through the embedding layers.

All sequences are ordered by default with numerical features first, followed by categorical features, as arranged in the datasets from the UCI Machine Learning Repository. For the ablation study, a dataset with 5,000 samples and 10 features—five numerical and five categorical—was simulated. The numerical features were generated with large correlations, including two pairs with correlations of 0.8 and 0.6, respectively. The categorical features were created with four distinct categories. Interaction terms were included as follows: An interaction between two numerical features, an interaction between a categorical and a numerical feature, and an interaction between two categorical features. The numerical features were scaled using standard normalization before generating the target variable. The target variable was constructed to include linear effects from each feature and the specified interaction terms, with added Gaussian noise for variability. We first fit a XGBoost model for a sanity check. Subsequently, we fit Mambular with default ordering (numerical before categorical features), flipped ordering and switched categorical and numerical ordering. Subsequently, we randomly shuffled the order and fit 10 models. We find that ordering does not have an effect on this simulated data, even with these large interaction and correlation effects[4].

---

[4]See the appendix for the chosen model parameters. Since the dataset is comparably smaller, we used a smaller Mambular model. Hyperparameters such as the learning rate, batch size etc. are kept identical to the default Mambular model.

Table 9: Performance for different orderings of features. Numerical features are given as integer numbers, categorical features as capital letters. Feature interaction between numerical features is given in blue. Feature interaction between categorical features is denoted in green and feature interaction between a numerical and a categorical feature is given in lavender. We find that reordering the features either before or after the embedding layers does not affect performance of the model. No ordering performs significantly better or worse than the default model, while all models perform significantly better than the XGBoost model.

| Before Embedding Layer | After Embedding Layer | Ordering |
|---|---|---|
| Default | $0.918 \pm 0.045$ | [1 2 3 4 5 A B C D E] |
| $0.916 \pm 0.043$ | $0.913 \pm 0.043$ | [E D C B A 5 4 3 2 1] |
| $0.919 \pm 0.044$ | $0.914 \pm 0.042$ | [A B C D E 1 2 3 4 5] |
| $0.917 \pm 0.043$ | $0.915 \pm 0.045$ | [A B 2 3 1 D E 4 C 5] |
| $0.920 \pm 0.046$ | $0.917 \pm 0.045$ | [D C 2 A B E 1 5 3 4] |
| $0.914 \pm 0.043$ | $0.914 \pm 0.044$ | [B 1 4 C D A 2 E 3 5] |
| $0.916 \pm 0.045$ | $0.914 \pm 0.041$ | [1 5 E B C 4 3 D 2 A] |
| $0.918 \pm 0.046$ | $0.914 \pm 0.045$ | [2 5 E B 4 A 1 3 D C] |
| $0.916 \pm 0.044$ | $0.915 \pm 0.043$ | [1 C A 2 D 4 E 3 5 B] |
| $0.917 \pm 0.040$ | $0.914 \pm 0.043$ | [A 1 4 5 2 C E B D 3] |
| $0.917 \pm 0.044$ | $0.922 \pm 0.040$ | [4 A 1 2 3 B 5 C D E] |
| $0.920 \pm 0.040$ | $0.913 \pm 0.040$ | [1 A D C B 3 E 2 5 4] |
| $0.920 \pm 0.041$ | $0.916 \pm 0.044$ | [C 5 B 2 4 A E D 3 1] |
| XGBoost | $1.096 \pm 0.038$ | |

## B.1 CALIFORNIA HOUSING

The p-values for the sequence ordering and positioning of Latitude and Longitude is given below.

Table 10: Detailed nalysis of results for CA Housing, including p-statistics.

| Model | CA $\downarrow$ | $p$-value | Ordering |
|---|---|---|---|
| Num\|Cat | $0.167 \pm 0.011$ | - | [LO, LA, MA, TR, TB, Po, Hh, MI, OP] |
| Cat\|Num | $0.158 \pm 0.007$ | 0.168 | [OP, MI, Hh, Po, TB, TR, MA, LA, LO] |
| | $0.177 \pm 0.007$ | 0.136 | [LO, MA, LA, TR, TB, Po, Hh, MI, OP] |
| | $0.175 \pm 0.008$ | 0.240 | [LO, MA, TR, LA, TB, Po, Hh, MI, OP] |
| | $\mathbf{0.194} \pm 0.010$ | 0.003 | [LO, MA, TR, TB, LA, Po, Hh, MI, OP] |
| | $\mathbf{0.196} \pm 0.011$ | 0.003 | [LO, MA, TR, TB, Po, LA, Hh, MI, OP] |
| | $\mathbf{0.194} \pm 0.011$ | 0.004 | [LO, MA, TR, TB, Po, Hh, LA, MI, OP] |
| | $\mathbf{0.195} \pm 0.010$ | 0.004 | [LO, MA, TR, TB, Po, Hh, MI, LA, OP] |
| | $\mathbf{0.194} \pm 0.012$ | 0.005 | [LO, MA, TR, TB, Po, Hh, MI, OP, LA] |

Given these results, and to verify, that the kernel size of 4 is the cause of this effect, we further analyzed the dataset. Below are more results for Mambular with random shuffling. Again we can see the the position of Latitude and Longitude significantly impact model performance, whenever these two variables are further apart than the fixed kernel size of 4.

To analyze the feature interaction effect between these two variables, we conducted a simple regression with pairwise feature interactions and analyzed the effect strengths. Interestingley, we find that the interaction between Longitude and Latitude is not as prominent as that between other variables.

Additionally, we have fit a XGboost model and analyzed the pairwise feature importance metrics and generally find the same results as for the linear regression.

Table 11: Analysis of results for CA Housing

| Model | CA ↓ | $p$-value | Ordering |
|---|---|---|---|
| Num\|Cat | $0.167 \pm 0.011$ | - | [LO, LA, MA, TR, TB, Po, Hh, MI, OP] |
| Cat\|Num | $0.158 \pm 0.007$ | 0.168 | [OP, MI, Hh, Po, TB, TR, MA, LA, LO] |
| | $0.174 \pm 0.009$ | 0.304 | [Po, Hh, MI, OP, LO, LA, MA, TR, TB] |
| | $\mathbf{0.195} \pm 0.012$ | 0.005 | [LO, MA, TR, TB, Po, Hh, MI, OP, LA] |
| | $\mathbf{0.197} \pm 0.010$ | 0.002 | [MA, LO, TR, TB, Po, Hh, MI, LA, OP] |
| | $\mathbf{0.188} \pm 0.010$ | 0.014 | [MA, TR, LO, TB, Po, Hh, LA, MI, OP] |
| | $0.178 \pm 0.010$ | 0.137 | [MA, LO, LA, TR, TB, Po, Hh, MI, OP] |
| | $0.177 \pm 0.008$ | 0.142 | [MA, TR, TB, Po, LA, LO, Hh, MI, OP] |
| | $0.178 \pm 0.009$ | 0.123 | [LA, LO, MA, TR, TB, Po, Hh, MI, OP] |
| | $0.172 \pm 0.070$ | 0.420 | [Hh, TB, Po, MI, MA, OP, LA, LO, TR] |
| | $0.177 \pm 0.010$ | 0.171 | [LO, Po, OP, LA, MI, MA, TR, Hh, TB] |
| | $\mathbf{0.190} \pm 0.010$ | 0.009 | [Hh, TB, LO, MI, Po, OP, TR, MA, LA] |

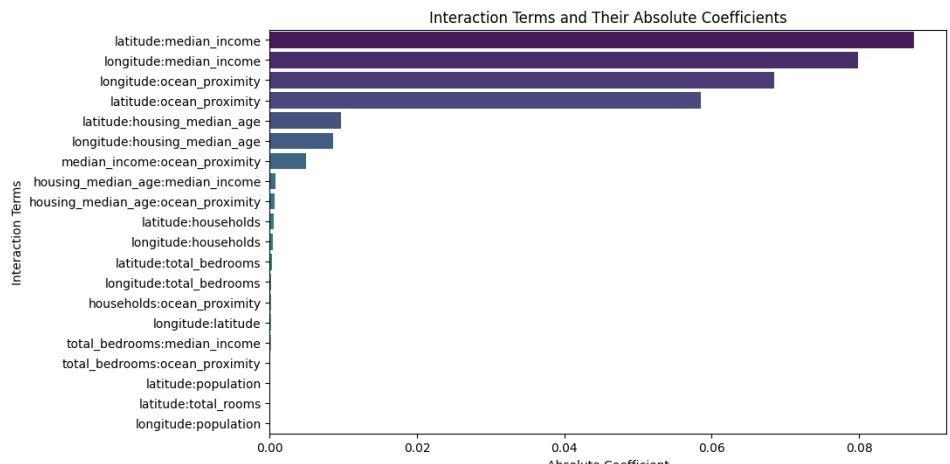

Figure 4: Linear Regression with pairwise interaction effects on the california housing dataset.

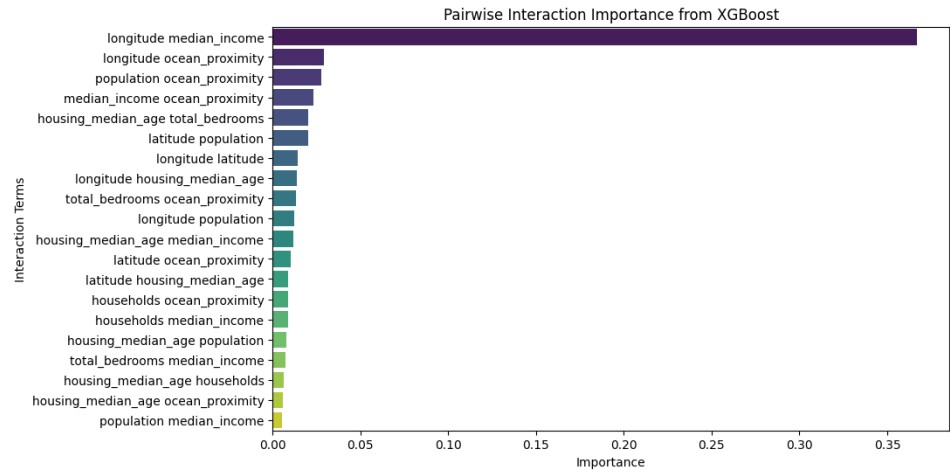

Figure 5: Pairwise feature importance statistics from a XGBoost model on the california housing dataset.

## C    DISTRIBUTIONAL REGRESSION

Distributional regression describes regression beyond the mean, i.e., the modeling of all distributional parameters. Thus, Location Scale and Shape (LSS) models can quantify the effects of covariates on not just the mean but also on any parameter of a potentially complex distribution assumed for the responses. A major advantage of these models is their ability to identify changes in all aspects of the response distribution, such as variance, skewness, and tail probabilities, enabling the model to properly disentangling aleatoric uncertainty from epistemic uncertainty.

This is achieved by minimizing the negative log-likelihood via optimizing the parameters $\theta$

$$\mathcal{L}(\theta) = -\sum_{i=1}^{n} \log f(y_i \mid \mathbf{x}_i, \theta)$$

For the two examples in the main part, a normal distribution is modelled and hence, the models minimize:

$$log\left(\mathcal{L}(\mu, \sigma^2 | y)\right) = -\frac{n}{2}\log(2\pi\sigma^2) - \frac{1}{2\sigma^2}\sum_{i=1}^{n}(y_i - \mu)^2,$$

where $n$ is the underlying number of observations and parameters $y \in \mathbb{R}$, location $\mu \in \mathbb{R}$ and scale $\sigma \in \mathbb{R}^+$.

While this has been a common standard in classical statistical approaches (Stasinopoulos and Rigby, 2008), it has not yet been widely adopted by the ML community. Recent interpretable approaches (Thielmann et al., 2024a), however, have demonstrated the applicability of distributional regression in tabular deep learning. Furthermore, approaches like XGBoostLSS (März, 2019; März and Kneib, 2022) demonstrate that tree-based models are capable of effectively solving such tasks. Below, we show that Mambular for Location Scale and Shape (MambularLSS) outperforms XGBoostLSS in terms of Continuous Ranked Probability Score (CRPS) (Gneiting and Raftery, 2007) when minimizing the negative log-likelihood while maintaining a small MSE.

**CRPS**    Analyzing distributional regression models also requires careful consideration of the evaluation metrics. Traditionally, mean focused models are evaluated using mean-centric metrics, e.g. MSE, AUC or Accuracy. However, a model that takes all distributional parameters into account should be evaluated on the predictive performance for all of the distributional parameters. Following Gneiting and Raftery (2007), the evaluation metric should be proper, i.e. enforce the analyst to report their true beliefs in terms of a predictive distribution. In terms of classical mean-centric metrics, e.g. MSE is proper for the mean, however, not proper for evaluating the complete distributional prediction. We therefore rely on the Continuous Ranked Probability Score (Gneiting and Raftery, 2007) for model evaluation, given by:

$$CRPS(F, x) = -\int_{-\infty}^{\infty} (F(y) - \mathbf{1}_{y \geq x})^2 \, dy.$$

See Gneiting and Raftery (2007) for more details.

Table 12: Results for distributional regression for a normal distribution for the Abalone and California Housing datasets. Significantly better models at the 5% level are marked in green. $p$-vales are 0.20 and 0.00002 respectively for Abalone and and CA housing for the CRPS metric.

| | AB | | CA | |
| | CRPS ↓ | MSE ↓ | CRPS ↓ | MSE ↓ |
|---|---|---|---|---|
| MambularLSS | $0.345 \pm 0.016$ | 0.458 | **0.201** $\pm 0.004$ | 0.181 |
| XGBoostLSS | $0.359 \pm 0.016$ | 0.479 | $0.227 \pm 0.005$ | 0.215 |

## D MAMBATAB

In addition to the popular tabular models described above, we tested the architecture proposed by Ahamed and Cheng (2024a). MambaTab is the first architecture to leverage Mamba blocks for tabular problems. However, the authors propose using a combined linear layer to project all inputs into a single feature representation, transforming the features into a pseudo-sequence of fixed length 1. This approach simplifies the recursive update from Eq. 2 into a matrix multiplication and makes the model resemble a ResNet due to the residual connections in the final processing. Utilizing a sequential model with a sequence length of 1 does not fully exploit the strengths of sequential processing, as it reduces the model's capacity to capture dependencies across multiple features.

We tested the architecture proposed by Ahamed and Cheng (2024a) and could achieve similar results for shared datasets, but overall found MambaTab to perform similar to a ResNet, aligning with expectations (see Table 2 and **??**). Additionally, we experimented with transposing the axes to create an input matrix of shape $(1) \times (\text{BATCH SIZE}) \times (\text{EMBEDDING DIMENSION})$, as outlined in their implementation. While this approach draws on ideas from TabPFN (Hollmann et al., 2022), it did not lead to performance improvements in our experiments. When using PLE encodings and increasing the number of layers and dimensions compared to the default implementation from Ahamed and Cheng (2024a) we are able to increase performance.

MambaTab (Ahamed and Cheng, 2024a) significantly differs from Mambular, since it is not a sequential model. To achieve the presented results from MambaTab, we have followed the provided implementation from the authors retrieved from `https://github.com/Atik-Ahamed/MambaTab`. It is worth noting, however, that MambaTab benchmarks the model on a lot of smaller datasets. 50% of the benchmarked datasets have not more than 1000 observations. Additionally, the provided implementation suggests, that MambaTab does indeed not iterate over a pseudo sequence length of 1, but rather over the number of observations, similar to a TabPFN (Hollmann et al., 2022). We have also tested this version, denoted as MambaTab$^T$ but did not find that it performs better than the described version. On the Adult dataset, our achieved result of 0.901 AUC on average is very similar to the default results reported in Ahamed and Cheng (2024a) with 0.906. The difference could be firstly due to us performing 5-fold cross validation and secondly different seeds in model initialization.

Table 13: Benchmarking results for the regression tasks for the original MambaTab implementation provided by `https://github.com/Atik-Ahamed/MambaTab`

| Models | DI ↓ | AB ↓ | CA ↓ | WI ↓ | PA ↓ | HS ↓ | CP ↓ |
|---|---|---|---|---|---|---|---|
| MambaTab | $0.035 \pm 0.006$ | $0.456 \pm 0.053$ | $0.272 \pm 0.016$ | $0.685 \pm 0.015$ | $0.531 \pm 0.032$ | $0.163 \pm 0.009$ | $0.030 \pm 0.002$ |
| MambaTab$^T$ | $0.038 \pm 0.002$ | $0.468 \pm 0.048$ | $0.279 \pm 0.010$ | $0.694 \pm 0.015$ | $0.576 \pm 0.022$ | $0.179 \pm 0.027$ | $0.033 \pm 0.002$ |

Table 14: Benchmarking results for the classification tasks. Average AUC values over 5 folds and the corresponding standard deviations are reported. Larger values are better.

| Models | BA ↑ | AD ↑ | CH ↑ | FI ↑ | MA ↑ |
|---|---|---|---|---|---|
| MambaTab | $0.886 \pm 0.006$ | $0.901 \pm 0.001$ | $0.828 \pm 0.005$ | $0.785 \pm 0.012$ | $0.880 \pm 0.003$ |
| MambaTab$^T$ | $0.888 \pm 0.005$ | $0.899 \pm 0.002$ | $0.815 \pm 0.009$ | $0.783 \pm 0.012$ | $0.878 \pm 0.005$ |

# E  DEFAULT MODEL HYPERPARAMETERS

In the following, we describe the default model parameters used for all the neural models. We based our choices on the literature to ensure meaningful and high-performing parameters by default. Additionally, we were able to reproduce results (on average) from popular studies, such as Gorishniy et al. (2021) and Grinsztajn et al. (2022). While most larger benchmark studies perform extensive hyperparameter tuning for each dataset, analyzing these results (Grinsztajn et al., 2022; Gorishniy et al., 2021) shows that most models already perform well without tuning, as also found by McElfresh et al. (2024). Furthermore, the results suggest that performing hyperparameter tuning for all models does not change the ranking between the models, since most models benefit from tuning to a similar degree. Thus, we have collected informed hyperparameter defaults which we list in the following. The hyperparameters such as learning rate, patience and number of epochs are shared among all models for a more consistent approach.

Table 15: Shared hyperparameters among all models

| Hyperparameter | Value |
| --- | --- |
| Learning rate | 1e-04 |
| Learning rate patience | 10 |
| Weight decay | 1e-06 |
| Learning rate factor | 0.1 |
| Max Epochs | 200 |

**MLP**   As a simple baseline, we fit a simple MLP without any special architecture. However, PLE encodings are used, as they have been shown to significantly improve performance.

Table 16: Default Hyperparameters for the MLP Model

| Hyperparameter | Value |
| --- | --- |
| Layer sizes | (256, 128, 32) |
| Activation function | SELU |
| Dropout rate | 0.5 |
| PLE encoding dimension | 128 |

**ResNet**   A ResNet architecture for tabular data has been shown to be a sensible baseline (Gorishniy et al., 2021). Furthermore, McElfresh et al. (2024) has validated the strong performance of ResNets compared to e.g. TabNet (Arik and Pfister, 2021) or NODE (Popov et al., 2019).

Table 17: Default Hyperparameters for the ResNet Model

| Hyperparameter | Value |
| --- | --- |
| Layer sizes | (256, 128, 32) |
| Activation function | SELU |
| Dropout rate | 0.5 |
| Skip connections | True |
| Batch normalization | True |
| Number of blocks | 3 |
| PLE encoding dimension | 128 |

**FT-Transformer**   For the FT-Transformer architecture we orientated on the default parameters conducted by Gorishniy et al. (2021). We only slightly adapted them from 3 layers and an embedding dimension of 192 to 4 layers and an embedding dimension of 128 to be more consistent with the other models. However, we tested out the exact same architecture from Gorishniy et al. (2021) and did not find any differences in performance, even a minimal (non-significant) decrease. Additionally,

we found that using ReGLU instead of ReLU activation function in the transformer blocks does improve performance consistently.

Table 18: Default Hyperparameters for the FT Transformer Model

| Hyperparameter | Value |
| --- | --- |
| Model Dim | 128 |
| Number of layers | 4 |
| Number of attention heads | 8 |
| Attention dropout rate | 0.2 |
| Feed-forward dropout rate | 0.1 |
| Normalization method | LayerNorm |
| Embedding activation function | Identity |
| Pooling method | cls |
| Normalization first in transformer block | False |
| Use bias in linear layers | True |
| Transformer activation function | ReGLU |
| Layer normalization epsilon | 1e-05 |
| Feed-forward layer dimensionality | 256 |
| PLE encoding dimension | 128 |

**TabTransformer**  We practically used the same hyperparameter for TabTransformer as we used for Ft-Transformer. For consistency we do not use a multi-layer MLP for where the contextualized embeddings are being passed to. While this deviates from the original architecture, leaving this out ensures a more consistent comparison to FT-Transformer and Mambular since both models use a single layer after pooling. However, we used a larger feed forward dimensionality in the transformer to counteract this. Overall, our results are in line with the literature and we can validate that Tab-Transformer outperforms a simple MLP on average. For datasets where no categorical features are available, the TabTransformer converges to a simple MLP. Thus we left these results blank in the benchmarks.

Table 19: Default Hyperparameters for the TabTransformer Model

| Hyperparameter | Value |
| --- | --- |
| Model Dim | 128 |
| Number of layers | 4 |
| Number of attention heads | 8 |
| Attention dropout rate | 0.2 |
| Feed-forward dropout rate | 0.1 |
| Normalization method | LayerNorm |
| Embedding activation function | Identity |
| Pooling method | cls |
| Normalization first in transformer block | False |
| Use bias in linear layers | True |
| Transformer activation function | ReGLU |
| Layer normalization epsilon | 1e-05 |
| Feed-forward layer dimensionality | 512 |
| PLE encoding dimension | 128 |

**MambaTab**  We test out three different MambaTab architectures. Firstly, we implement the same architecture as for Mambular but instead of an embedding layer for each feature and creating a sequence of length $J$ we feed all features jointly through a single embedding layer and create a sequence of length 1. The *Axis* argument thus specifies over which axis the SSM model iterates. As described by Ahamed and Cheng (2024a) the model iterates over this pseudo-sequence length of 1.

Additionally, we test out the default architecture from Ahamed and Cheng (2024a) and hence have a super small model with only a single layer and embedding dimensionality of 32.

Table 20: Default Hyperparameters for the MambaTab* Model

| Hyperparameter | Value |
| --- | --- |
| Model Dim | 64 |
| Number of layers | 4 |
| Expansion factor | 2 |
| Kernel size | 4 |
| Use bias in convolutional layers | True |
| Dropout rate | 0.0 |
| Dimensionality of the state | 128 |
| Normalization method | RMSNorm |
| Activation function | SiLU |
| PLE encoding dimension | 64 |
| Axis | 1 |

Table 21: Default Hyperparameters for the MambaTab Model

| Hyperparameter | Value |
| --- | --- |
| Model Dim | 32 |
| Number of layers | 1 |
| Expansion factor | 2 |
| Kernel size | 4 |
| Use bias in convolutional layers | True |
| Dropout rate | 0.0 |
| Dimensionality of the state | 32 |
| Normalization method | RMSNorm |
| Activation function | SiLU |
| Axis | 1 |

Lastly, we follow the Github implementation from Ahamed and Cheng (2024a) where the sequence is flipped and the SSM iterates over the number of observations instead of the pseudo-sequence length of 1.

Table 22: Default Hyperparameters for the MambaTab$^T$ Model

| Hyperparameter | Value |
| --- | --- |
| Model Dim | 32 |
| Number of layers | 1 |
| Expansion factor | 2 |
| Kernel size | 4 |
| Use bias in convolutional layers | True |
| Dropout rate | 0.0 |
| Dimensionality of the state | 32 |
| Normalization method | RMSNorm |
| Activation function | SiLU |
| Axis | 0 |

**Mambular** For Mambular we create a sensible default, following hyperparameters from the literature. We keep all hyperparameters from the Mambablocks as introduced by Gu and Dao (2023). Hence we use SiLU activation and RMSNorm. WE use an expansion factor of 2 and use an embedding dimensionality of 64. The PLE encoding dimension is adapted to always match the embedding dimensionalitiy since first expanding the dimensionality in preprocessing to subsequently reduce it in the embedding layer seems counter intuitive.

Table 23: Default Hyperparameters for the Mambular Model

| Hyperparameter | Value |
|---|---|
| Model Dim | 64 |
| Number of layers | 4 |
| Expansion factor | 2 |
| Kernel size | 4 |
| Use bias in convolutional layers | True |
| Dropout rate | 0.0 |
| Dimensionality of the state | 128 |
| Normalization method | RMSNorm |
| Activation function | SiLU |
| PLE encoding dimension | 64 |

**Model sizes** Below you find the number of trainable parameters for all models for all datasets. Note, that MambaTab$^*$ and Mambular have very similar numbers of parameters since the sequence length does not have a large impact on the number of model parameters. Overall there is no correlation between model size and performance since e.g. the FT-Transformer architecture which is comparably larger to e.g. the MLP and ResNet architectures performs very well whereas the largest architecture, the TabTransformer performs worse than the smaller ResNet. Additionally, since the models have distinctively different architectures, the overall number of trainable parameters is not conclusive for training time or memory usage.

Table 24: Number of trainable parameters for all models and datasets. Note that the number of trainable parameters is dependent on the dataset, since e.g. a larger number of variables leads to more trainable parameters in the embedding layer.

| Dataset | AB | AD | BA | CA | CH | CP | DI | FI | HS | MA | PA | WI |
|---|---|---|---|---|---|---|---|---|---|---|---|---|
| FT-Transformer | 765k | 709k | 795k | 784k | 722k | 852k | 763k | 834k | 837k | 794k | 944k | 822k |
| MLP | 242k | 103k | 124k | 280k | 156k | 418k | 233k | 351k | 310k | 105k | 594k | 356k |
| ResNet | 261k | 123k | 144k | 299k | 176k | 437k | 253k | 371k | 330k | 125k | 614k | 375k |
| TabTransformer | 1060k | 1073k | 1149k | 1061k | 1060k | - | 1063k | - | 1100k | 1157k | 1068k | - |
| MambaTab$^*$ | 331k | 318k | 316k | 335k | 321k | 352k | 328k | 358k | 339k | 312k | 373k | 348k |
| MambaTab | 13k | 14k | 14k | 13k | 13k | 14k | 13k | 14k | 14k | 14k | 14k | 14k |
| Mambular | 331k | 324k | 361k | 335k | 321k | 352k | 329k | 365k | 358k | 361k | 374k | 348k |

# F  RESULTS

All model performances, including the MambaTab variants are given below.

Table 25: Benchmarking results for the regression tasks. Average mean squared error values over 5 folds and the corresponding standard deviations are reported. Smaller values are better. The best performing model is marked in bold.

| Models | DI $\downarrow$ | AB $\downarrow$ | CA $\downarrow$ | WI $\downarrow$ | PA $\downarrow$ | HS $\downarrow$ | CP $\downarrow$ |
|---|---|---|---|---|---|---|---|
| XGBoost | $0.019 \pm 0.000$ | $0.506 \pm 0.044$ | $0.171 \pm 0.007$ | $0.528 \pm 0.008$ | $0.036 \pm 0.004$ | $0.119 \pm 0.024$ | $0.024 \pm 0.004$ |
| RF | $0.019 \pm 0.001$ | $0.461 \pm 0.052$ | $0.183 \pm 0.008$ | $\mathbf{0.485} \pm 0.007$ | $0.028 \pm 0.006$ | $0.121 \pm 0.018$ | $0.025 \pm 0.002$ |
| LightGBM | $0.019 \pm 0.001$ | $0.459 \pm 0.047$ | $0.171 \pm 0.007$ | $0.542 \pm 0.013$ | $0.039 \pm 0.007$ | $0.112 \pm 0.018$ | $0.023 \pm 0.003$ |
| CatBoost | $0.019 \pm 0.000$ | $0.457 \pm 0.007$ | $0.169 \pm 0.006$ | $0.583 \pm 0.006$ | $0.045 \pm 0.006$ | $\mathbf{0.106} \pm 0.015$ | $\mathbf{0.022} \pm 0.001$ |
| FT-Transformer | $0.018 \pm 0.001$ | $0.458 \pm 0.055$ | $0.169 \pm 0.006$ | $0.615 \pm 0.012$ | $\mathbf{0.024} \pm 0.005$ | $0.111 \pm 0.014$ | $0.024 \pm 0.001$ |
| MLP | $0.066 \pm 0.003$ | $0.462 \pm 0.051$ | $0.198 \pm 0.011$ | $0.654 \pm 0.013$ | $0.764 \pm 0.023$ | $0.147 \pm 0.017$ | $0.031 \pm 0.001$ |
| TabTransformer | $0.065 \pm 0.002$ | $0.472 \pm 0.057$ | $0.247 \pm 0.013$ | - | $0.135 \pm 0.001$ | $0.160 \pm 0.028$ | - |
| ResNet | $0.039 \pm 0.018$ | $0.455 \pm 0.045$ | $0.178 \pm 0.006$ | $0.639 \pm 0.013$ | $0.606 \pm 0.031$ | $0.141 \pm 0.017$ | $0.030 \pm 0.002$ |
| NODE | $0.019 \pm 0.000$ | $\mathbf{0.431} \pm 0.052$ | $0.207 \pm 0.001$ | $0.613 \pm 0.006$ | $0.045 \pm 0.007$ | $0.124 \pm 0.015$ | $0.026 \pm 0.001$ |
| LinReg | $0.115 \pm 0.002$ | $0.483 \pm 0.055$ | $0.365 \pm 0.021$ | $0.711 \pm 0.006$ | $0.830 \pm 0.047$ | $0.302 \pm 0.033$ | $0.289 \pm 0.004$ |
| MambaTab | $0.035 \pm 0.006$ | $0.456 \pm 0.053$ | $0.272 \pm 0.016$ | $0.685 \pm 0.015$ | $0.531 \pm 0.032$ | $0.163 \pm 0.009$ | $0.030 \pm 0.002$ |
| MambaTab$^T$ | $0.038 \pm 0.002$ | $0.468 \pm 0.048$ | $0.279 \pm 0.010$ | $0.694 \pm 0.015$ | $0.576 \pm 0.022$ | $0.179 \pm 0.027$ | $0.033 \pm 0.002$ |
| MambaTab$^*$ | $0.040 \pm 0.008$ | $0.455 \pm 0.043$ | $0.180 \pm 0.008$ | $0.601 \pm 0.010$ | $0.571 \pm 0.021$ | $0.122 \pm 0.017$ | $0.030 \pm 0.002$ |
| Mambular | $\mathbf{0.018} \pm 0.000$ | $0.452 \pm 0.043$ | $\mathbf{0.167} \pm 0.011$ | $0.628 \pm 0.010$ | $0.035 \pm 0.005$ | $0.132 \pm 0.020$ | $0.025 \pm 0.002$ |

Table 26: Benchmarking results for the classification tasks. Average AUC values over 5 folds and the corresponding standard deviations are reported. Larger values are better.

| Models | BA ↑ | AD ↑ | CH ↑ | FI ↑ | MA ↑ |
|---|---|---|---|---|---|
| XGBoost | $0.928 \pm 0.004$ | $\mathbf{0.929} \pm 0.002$ | $0.845 \pm 0.008$ | $0.774 \pm 0.009$ | $0.922 \pm 0.002$ |
| RF | $0.923 \pm 0.006$ | $0.896 \pm 0.002$ | $0.851 \pm 0.008$ | $0.789 \pm 0.012$ | $0.917 \pm 0.004$ |
| LightGBM | $\mathbf{0.932} \pm 0.004$ | $0.929 \pm 0.001$ | $0.861 \pm 0.008$ | $0.788 \pm 0.010$ | $\mathbf{0.927} \pm 0.001$ |
| CatBoost | $0.932 \pm 0.008$ | $0.927 \pm 0.002$ | $\mathbf{0.867} \pm 0.006$ | $0.796 \pm 0.010$ | $0.926 \pm 0.005$ |
| FT-Transformer | $0.926 \pm 0.003$ | $0.926 \pm 0.002$ | $0.863 \pm 0.007$ | $0.792 \pm 0.011$ | $0.916 \pm 0.003$ |
| MLP | $0.895 \pm 0.004$ | $0.914 \pm 0.002$ | $0.840 \pm 0.005$ | $0.793 \pm 0.011$ | $0.886 \pm 0.003$ |
| TabTransformer | $0.921 \pm 0.004$ | $0.912 \pm 0.002$ | $0.835 \pm 0.007$ | - | $0.910 \pm 0.002$ |
| ResNet | $0.896 \pm 0.006$ | $0.917 \pm 0.002$ | $0.841 \pm 0.006$ | $0.793 \pm 0.013$ | $0.889 \pm 0.003$ |
| NODE | $0.914 \pm 0.008$ | $0.904 \pm 0.002$ | $0.851 \pm 0.006$ | $0.790 \pm 0.010$ | $0.904 \pm 0.005$ |
| Log-Reg | $0.810 \pm 0.008$ | $0.838 \pm 0.001$ | $0.754 \pm 0.006$ | $0.768 \pm 0.013$ | $0.800 \pm 0.005$ |
| MambaTab* | $0.900 \pm 0.004$ | $0.916 \pm 0.003$ | $0.846 \pm 0.007$ | $0.792 \pm 0.011$ | $0.890 \pm 0.003$ |
| MambaTab | $0.886 \pm 0.006$ | $0.901 \pm 0.001$ | $0.828 \pm 0.005$ | $0.785 \pm 0.012$ | $0.880 \pm 0.003$ |
| MambaTab$^T$ | $0.888 \pm 0.005$ | $0.899 \pm 0.002$ | $0.815 \pm 0.009$ | $0.783 \pm 0.012$ | $0.878 \pm 0.005$ |
| Mambular | $0.927 \pm 0.006$ | $0.928 \pm 0.002$ | $0.861 \pm 0.008$ | $\mathbf{0.796} \pm 0.013$ | $0.917 \pm 0.003$ |

Table 27: Combined Ranking of Models for Regression and Classification Tasks

| Models | Regression Rank | Classification Rank | Overall Rank |
|---|---|---|---|
| XGBoost | $5.14 \pm 4.02$ | $5.4 \pm 4.51$ | $5.25 \pm 4.03$ |
| RF | $5.00 \pm 2.94$ | $7.4 \pm 3.36$ | $6.00 \pm 3.22$ |
| LightGBM | $4.57 \pm 2.07$ | $3.4 \pm 3.36$ | $4.08 \pm 2.61$ |
| CatBoost | $4.00 \pm 2.45$ | $\mathbf{2.2} \pm 1.10$ | $\mathbf{3.25} \pm 2.14$ |
| FT-Transformer | $\mathbf{3.57} \pm 2.51$ | $4.6 \pm 1.52$ | $4.00 \pm 2.13$ |
| MLP | $10.86 \pm 1.57$ | $8.6 \pm 3.36$ | $9.92 \pm 2.61$ |
| TabTransformer | $10.80 \pm 1.64$ | $8.5 \pm 1.91$ | $9.78 \pm 2.05$ |
| ResNet | $8.14 \pm 2.91$ | $7.6 \pm 3.05$ | $7.92 \pm 2.84$ |
| NODE | $5.71 \pm 2.93$ | $7.6 \pm 1.82$ | $6.50 \pm 2.61$ |
| Regression | $13.57 \pm 0.53$ | $13.8 \pm 0.45$ | $13.67 \pm 0.49$ |
| MambaTab | $9.29 \pm 2.56$ | $11.6 \pm 1.14$ | $10.25 \pm 2.34$ |
| MambaTab$^T$ | $11.57 \pm 1.40$ | $12.2 \pm 0.84$ | $11.83 \pm 1.19$ |
| MambaTab* | $7.11 \pm 2.79$ | $7.4 \pm 1.67$ | $7.25 \pm 2.30$ |
| Mambular | $4.00 \pm 3.06$ | $3.0 \pm 1.22$ | $3.58 \pm 2.43$ |

Further results on a regression benchmark with a single train-test-validation split are reported below. The datasets are taken from Fischer et al. (2023) with datasets already present in the main results excluded.

Table 28: Comparison of models on an additional regression benchmark. Mambular and CatBoost perform best among compared models

| Model | BH ↓ | CW ↓ | FF ↓ | GS ↓ | HI ↓ | K8 ↓ | AV ↓ | KC ↓ | MH ↓ | NP ↓ | PP ↓ | SA ↓ | SG ↓ | VT ↓ | Rank ↓ |
|---|---|---|---|---|---|---|---|---|---|---|---|---|---|---|---|
| Mambular | **0.021** | **0.701** | 0.272 | 0.057 | **0.595** | 0.168 | 0.018 | 0.137 | 0.085 | **0.003** | 0.402 | **0.015** | 0.318 | **0.003** | **1.79** |
| FTTransformer | 0.028 | 0.701 | 0.301 | 0.205 | 0.609 | 0.451 | 0.089 | 0.149 | 0.101 | 0.009 | 0.542 | 0.033 | 0.360 | 0.045 | 4.36 |
| CatBoost | 0.032 | 0.702 | **0.245** | **0.041** | 0.597 | **0.150** | 0.004 | **0.110** | **0.078** | 0.005 | **0.390** | 0.018 | **0.297** | 0.013 | **1.79** |
| LightGBM | 0.048 | 0.707 | 0.263 | 0.059 | 0.599 | 0.239 | 0.024 | 0.140 | 0.091 | 0.009 | 0.452 | 0.031 | 0.302 | 0.013 | 3.26 |
| XGBoost | 0.039 | 0.752 | 0.281 | 0.078 | 0.635 | 0.259 | **0.004** | 0.161 | 0.098 | 0.006 | 0.403 | 0.024 | 0.329 | 0.013 | 3.71 |

