# OpenReview forum: "Mambular: A Sequential Model for Tabular Deep Learning"
_ICLR.cc/2025/Conference — Submitted to ICLR 2025_

### Official Review · Reviewer_Na6p · 2024-10-31

**Soundness:** 1
**Presentation:** 2
**Contribution:** 1
**Rating:** 3
**Confidence:** 4

**Summary:**

The paper introduces Mambular, an adaptation of the Mamba architecture specifically designed for tabular data problems. The authors claim that their model leverages a sequential interpretation of tabular data, similar to its successful application in other domains such as text, vision, and time series. The proposed architecture incorporates pooling strategies and feature interaction mechanisms to enhance performance. The paper provides experimental results on various datasets, comparing Mambular with state-of-the-art models such as FT-Transformer, TabTransformer, XGBoost, and LightGBM. According to the authors, Mambular achieves competitive results, highlighting the feasibility of applying sequential models to tabular data.

**Strengths:**

The paper explores the application of the Mamba architecture to tabular data, leveraging lessons from its success in handling other data types like text and time series. While not entirely novel, this extension demonstrates an attempt to generalize Mamba to a new domain.

The authors present experimental results comparing Mambular with a limited selection of state-of-the-art models, such as FT-Transformer, TabTransformer, XGBoost, and LightGBM. While the comparisons are not exhaustive, these initial results provide some insight into the model’s potential.

The architecture incorporates feature interaction mechanisms and various pooling strategies, potentially providing flexibility to handle different types of tabular data characteristics.

**Weaknesses:**

- Lack of novelty: One of the main weaknesses is the lack of significant novelty. The concept of adapting Mamba for tabular data is not substantially different from previously proposed models like MambaTab [1]. The minor modifications presented in Mambular mainly involve hyperparameter tuning and slight architectural changes, which do not justify the need for a new model.

[1] MA Ahamed et al., MambaTab: A Plug-and-Play Model for Learning Tabular Data, MIPR, 2024.

- Misalignment with tabular data characteristics: The paper fails to convincingly justify why treating tabular data as sequential offers any distinct advantage. Unlike sequential data such as text or time series, tabular data lacks a natural ordering of features. The authors did not provide clear evidence or theoretical grounding for why a sequence-based model should work better in this context.

- Limited scope of experiments: The experimental results are based on only 12 datasets, and the comparison to state-of-the-art algorithms is not comprehensive. Given the varied nature of tabular data, a broader set of datasets and comparisons to more established tabular models would have strengthened the claims.

- Inconsistent Performance: Despite the claims of Mambular’s superiority, the reported results do not consistently show a significant performance improvement over existing methods, such as XGBoost or LightGBM.

**Questions:**

What is the rationale behind interpreting tabular data as sequences? Could the authors provide a theoretical explanation or empirical evidence showing that treating features as a sequence offers a distinct advantage?

How do the authors address the lack of consistency in performance improvements across the datasets? Were any specific types of datasets or characteristics where Mambular performed particularly well?

Could the authors explain why comparisons were not made with a wider range of established tabular models, such as TabR[2], T2G-Former[3], and SAINT[4]?
[2] TabR: Unlocking the Power of Retrieval-Augmented Tabular Deep Learning, 2023
[3] T2G-Former: Organizing Tabular Features into Relation Graphs Promotes Heterogeneous Feature Interaction, 2023
[4] SAINT: Improved Neural Networks for Tabular Data via Row Attention and Contrastive Pre-Training, 2021

---

> ### Author Response · Authors · 2024-11-20
> **Answer to Comments/Questions**
>
> Dear Reviewer, thank you for your comments. We address your points below.
>
> > Lack of novelty: One of the main weaknesses is the lack of significant novelty. The concept of adapting Mamba for tabular data is not substantially different from previously proposed models like MambaTab [1]. The minor modifications presented in Mambular mainly involve hyperparameter tuning and slight architectural changes, which do not justify the need for a new model.
>
> - Thank you for this comment. We would like to clarify the comparison to MambaTab [1]. MambaTab is **not** a sequential model; rather, it is more similar to a ResNet architecture, as it embeds all features jointly into a single layer, resulting in a single embedding of shape (Batch Size) x (1) x (Embedding Dimension). Here, (1) typically represents sequence length, or in Mambular's case, the number of features. In MambaTab, however, all features are fused into a single representation, unlike Mambular, which treats each feature individually in sequence.
>
> - We encountered several issues with both the MambaTab paper and code. In the paper, MambaTab processes all features through a single layer, resulting in an input of (Batch Size) x (1) x (Embedding Dimension). This structure renders the SSM step ineffective, as there is no sequence to iterate over. However, upon analyzing their code, we found that it actually uses an approach more like (1) x (Batch Size) x (Embedding Dimension), iterating over observations rather than features, which more closely resembles the architecture in [2]. We conducted extensive comparisons with both interpretations in the appendix and found MambaTab to perform significantly worse than Mambular. However, due to these discrepancies in both the paper and code, we refrained from including these results in the main text. We would appreciate your guidance on how best to proceed here.
>
> > Misalignment with tabular data characteristics: The paper fails to convincingly justify why treating tabular data as sequential offers any distinct advantage. Unlike sequential data such as text or time series, tabular data lacks a natural ordering of features. The authors did not provide clear evidence or theoretical grounding for why a sequence-based model should work better in this context.
>
> - This remark is somewhat surprising, as this is a central contribution of our work. To the best of our knowledge, Mambular is the first model to effectively demonstrate that tabular problems can be accurately addressed using a sequential model. Although tabular data lacks natural feature ordering, our results show that sequential models, which update predictions feature-by-feature, can be a viable and high-performing alternative. This finding is novel in itself and contributes new insights into modeling tabular data.
>
> > Limited scope of experiments: The experimental results are based on only 12 datasets, and the comparison to state-of-the-art algorithms is not comprehensive. Given the varied nature of tabular data, a broader set of datasets and comparisons to more established tabular models would have strengthened the claims.
>
> - Please see our general response for additional details on this.
>
> > Inconsistent Performance: Despite the claims of Mambular’s superiority, the reported results do not consistently show a significant performance improvement over existing methods, such as XGBoost or LightGBM.
>
> - As clearly stated in our paper, Mambular performs best on average across our benchmarks. However, as with any model, certain datasets favor other approaches. This aligns well with existing literature; expecting a model—especially one with a new fitting paradigm—to outperform well-established benchmarks across all datasets would be unrealistic. See [1], where each model, even the top-performing model on average, performs poorly on specific tasks.

---

> > ### Author Response · Authors · 2024-11-20
> > **Answer Contd.**
> >
> > > What is the rationale behind interpreting tabular data as sequences? Could the authors provide a theoretical explanation or empirical evidence showing that treating features as a sequence offers a distinct advantage?
> >
> > - Viewing tabular data as a sequence enables feature-incremental learning [3], among other benefits. Additionally, using Mamba instead of Transformer architectures allows Mambular to handle large datasets with thousands of features without encountering out-of-memory issues.
> >
> > > How do the authors address the lack of consistency in performance improvements across the datasets? Were any specific types of datasets or characteristics where Mambular performed particularly well?
> >
> > - Mambular actually performs quite consistently. While it does not outperform on every single task, it ranks best on average for classification tasks and second-best on average for regression tasks. The only dataset where it struggled notably was the House Sales dataset, where most DL models except for FT-Transformer had difficulties. Additionally, Mambular has the lowest standard deviation in rankings (aside from the underperforming TabTransformer).
> >
> > > Could the authors explain why comparisons were not made with a wider range of established tabular models, such as TabR, T2G-Former, and SAINT?
> >
> > - Please refer to our general response for a detailed explanation on this point.
> >
> > ---
> >
> > Thank you once again for your thoughtful feedback. We hope these responses provide clarification and reinforce our contributions.
> >
> > ---
> > [1] McElfresh, Duncan, et al. "When do neural nets outperform boosted trees on tabular data?."
> > [2] Popov, Sergei, et al. "Neural oblivious decision ensembles for deep learning on tabular data."
> > [3] Van de Ven, Gido M., et al. "Three types of incremental learning." Nature Machine Intelligence (2022)

---

> > > ### Author Response · Authors · 2024-11-24
> > > **Rebuttal Reminder**
> > >
> > > Dear Reviewer,
> > >
> > > as the rebuttal period ends soon, we hope our responses to your review have addressed your concerns. We’d greatly value your input to ensure the manuscript fully reflects your expectations - please let us know if there are additional points we can clarify or refine.

---

> > > > ### Author Response · Authors · 2024-11-30
> > > > **Invitation to start the discussion**
> > > >
> > > > Dear Reviewer,
> > > >
> > > > We would like to kindly remind you that the extended rebuttal period will end in less than 72 hours.
> > > >
> > > > We have done our best to address your concerns and answer your questions. In summary, we have:
> > > > - Extended our benchmarks with multiple publicly available datasets.
> > > > - Included several new models.
> > > > - Answered questions regarding novelty and distinguished Mambular from MambaTab.
> > > >
> > > > We also kindly draw your attention to our general response, which addresses many of your initial concerns and questions.
> > > > We would greatly appreciate it if you could let us know if you have any remaining concerns. If not, we kindly ask you to consider reflecting these changes in your scores.
> > > >
> > > > We look forward to your feedback and to starting the discussion.

---

### Official Review · Reviewer_7hmN · 2024-11-02

**Soundness:** 3
**Presentation:** 4
**Contribution:** 3
**Rating:** 8
**Confidence:** 5

**Summary:**

This paper presents Mambular, a sequential deep learning model based on the Mamba architecture, designed for tabular data tasks. The authors evaluate Mambular against state-of-the-art neural and (ensemble) tree-based models, showing that it performs as well as or better than these models across different datasets.

**Strengths:**

- A novel sequential approach to tabular data, treating features as ordered sequences to capture feature interactions.

- Solid benchmarks against state-of-the-art models confirm its competitive performance across diverse datasets.

- Paper is clearly written and easy to follow,  the model and its components are clearly explained, and the comprehensive ablation study provides insight into the impact of different architectural choices.

**Weaknesses:**

- The paper would benefit from a more extensive comparison with recent deep learning baselines for tabular data. While it presents strong benchmarks, adding models like TabPFN and GANDALF could enhance the evaluation, giving a clearer picture of Mambular’s performance against the latest advancements.
- Additionally, thorough hyperparameter tuning is necessary for many tabular learning models to ensure fair and optimal comparisons. Please refer to Questions.

**Questions:**

- Hyperparameter Tuning and Model Performance: In Section 5, the authors claim that hyperparameter tuning does not significantly impact model ranking, citing Grinsztajn et al. (2022) and Gorishniy et al. (2021). I find this very questionable, especially given that Table 3 shows Random Forest outperforming XGBoost, which is typically strong for tabular data (Also, see Appendix tables here [1]).
Additionally, the model’s sensitivity to kernel size in permutation experiments suggests that hyperparameter choices affect performance significantly. To address this, I suggest comparing models using an established benchmark such as TabZilla [2], which offers a diverse suite of datasets for evaluating tabular models.

- The Mambular model’s sequential structure makes it highly sensitive to feature order, which impacts performance under certain hyperparameter settings. Although feature order dependency is a known issue in tabular data models, I recommend that the authors conduct an experiment with random feature permutations (see Definition 2 in [3]). This could enable Mambular to emphasize dependencies rather than sequence, potentially improving model's robustness.



[1] https://arxiv.org/abs/2305.13072

[2] https://github.com/naszilla/tabzilla

[3] Borisov, V., Seßler, K., Leemann, T., Pawelczyk, M., & Kasneci, G. (2022). "Language models are realistic tabular data generators." arXiv preprint arXiv:2210.06280.

---

> ### Author Response · Authors · 2024-11-20
> **Answer to Comments/Questions**
>
> Dear Reviewer,
> Thank you for your thoughtful and constructive feedback on our work. We appreciate the opportunity to address your comments and suggestions.
>
> ## Weaknesses
>
> > The paper would benefit from a more extensive comparison with recent deep learning baselines for tabular data. While it presents strong benchmarks, adding models like TabPFN and GANDALF could enhance the evaluation, giving a clearer picture of Mambular’s performance against the latest advancements.
>
> - Please refer to our general response for additional context on this point. Unfortunately, all our datasets are beyond the scope of TabPFN, which additionally supports only classification tasks and not regression.
>
> - As an initial step to broaden model comparisons, we would like to refer you to our general response, where we included additional models as well as 15 additional tasks. All results are included in the revised version of our manuscript.
>
> > Hyperparameter Tuning and Model Performance: In Section 5, the authors claim that hyperparameter tuning does not significantly impact model ranking, citing Grinsztajn et al. (2022) and Gorishniy et al. (2021). I find this very questionable, especially given that Table 3 shows Random Forest outperforming XGBoost, which is typically strong for tabular data (Also, see Appendix tables here [1]).
>
> - Please note that our claim refers to **average rankings**. While individual models may show performance gains with hyperparameter optimization, [2, 3, 4] all suggest that the relative ranking between models remains stable, as hyperparameter optimization does not substantially alter the average rankings across datasets.
>
> - Regarding Random Forest, it is indeed shown in Table 2 to be more than a full rank below XGBoost, aligning with expectations. While XGBoost generally outperforms Random Forest on average, there are specific datasets for which Random Forest can perform better, as noted in [2].
>
> > Additionally, the model’s sensitivity to kernel size in permutation experiments suggests that hyperparameter choices affect performance significantly. To address this, I suggest comparing models using an established benchmark such as TabZilla [2], which offers a diverse suite of datasets for evaluating tabular models.
>
> - Please see the analysis in the Appendix for this. While kernel size affects performance in cases of strong feature interactions, we show that setting the kernel size equal to the number of features fully mitigates this issue, as outlined in both Section 5 and the Appendix. Additionay, this only supports our argument that HPO would likely further improve Mambular's performance relative to the baselines, as the baselines already benefit from well-established default parameters, while Mambular has not yet been optimized to the same extent.
>
> - Please also see our general response regarding the benchmarks in [2].
>
> > Mambular's sequential structure makes it highly sensitive to feature order, which impacts performance under certain hyperparameter settings. Although feature order dependency is a known issue in tabular data models, I recommend that the authors conduct an experiment with random feature permutations (see Definition 2 in [3]). This could enable Mambular to emphasize dependencies rather than sequence, potentially improving model's robustness.
>
> - We kindly ask you to review the relevant sections again, as we indeed conduct feature permutation experiments and discuss our findings. Specifically, we show that setting the kernel size to the number of features mitigates issues associated with feature order. This solution, as detailed in Section 5 and the Appendix, effectively enhances Mambular’s robustness without compromising performance.
>
> ---
>
> Thank you once again for your thoughtful feedback. We hope these responses provide clarification and reinforce our contributions.
>
>
> ---
> [1] Gorishniy, Yury, et al. "TabM: Advancing Tabular Deep Learning with Parameter-Efficient Ensembling."
> [2] McElfresh, Duncan, et al. "When do neural nets outperform boosted trees on tabular data?."
> [3] Gorishniy, Yury, et al. "Revisiting deep learning models for tabular data."

---

> > ### Comment · Reviewer_7hmN · 2024-11-20
> >
> > Thank you for your response. I’ve raised my rating as most of my concerns have been addressed.
> >
> > Regarding the default hyperparameters, "average rankings" feels too vague to specify, but this is another topic and it is unrelated to the paper.

---

### Official Review · Reviewer_2WJ3 · 2024-11-03

**Soundness:** 2
**Presentation:** 2
**Contribution:** 2
**Rating:** 3
**Confidence:** 3

**Summary:**

The authors introduced Mamba architecture to solve the problems of tabular data. They showed the effectiveness of the proposed method by comparing its performance with neural networks-based algorithms and tree-based methods. And they confirmed the superiority of a sequence and passing method by comparing various pooling methods.

**Strengths:**

The authors proposed the Mamba architecture to solve tabular problems. It showed good performance overall, especially in classification performance.

**Weaknesses:**

The proposed method seems to be nothing more than using the Mamba structure on tabular data. Although there are some structural suggestions, it is difficult to say that the idea is novel overall.
And when I check the experimental results, it seems difficult to say that the performance has been greatly improved.Additional discussions and experiments are needed to see what advantages can be taken advantage of by applying the Mamba structure to tabular data, what differential performance improvements can be made compared to existing methods, and what can be considered if a Mamba structure or a new structure based on Mamba is proposed in the future.

**Questions:**

Please check the above weaknesses.

---

> ### Author Response · Authors · 2024-11-20
> **Response to Questions and Weaknesses**
>
> Dear Reviewer,
>
> Thank you for your thoughtful and constructive feedback on our work. We greatly appreciate the time and effort you have invested in providing valuable insights. Below, we have carefully addressed your comments and answered your questions in detail.
> Additionally, we have given a detailed general response above.
>
> > ... it is difficult to say that the idea is novel overall.
>
> - Please see our general response regarding this point.
>
> > And when I check the experimental results, it seems difficult to say that the performance has been greatly improved.
>
> - Thank you for this comment. We would firstly like to refer you to our general response.
>
> -  Second, we would like to emphasize that Mambular, without any tuning or expert-selected hyperparameters (unlike baselines such as XGBoost and FT-Transformer), performs extremely strong and even oputperforms proven Baselines such as LightGBM. Given the fundamentally different approach Mambular takes — treating tabular data as a sequence — this is remarkable. In the tabular domain, where benchmark improvements are modest, even marginal advances are important.
>
>
> > Be taken advantage of by applying the Mamba structure to tabular data, what differential performance improvements can be made compared to existing methods, and what can be considered if a Mamba structure or a new structure based on Mamba is proposed in the future.
>
> - Thank you for this thoughtful question. Firstly, we direct you to the appendix, where we present some additional experiments, including distributional regression and feature ordering. Secondly, we emphasize Mambular's efficiency. Using Mambalayers makes Mambular much more memory efficient than Transformer based architectures.
>
> ---
>
> We hope these responses clarify the contributions and performance of Mambular and provide further insights into its advantages and potential applications. Thank you once again for your review.

---

> > ### Author Response · Authors · 2024-11-24
> > **Rebuttal Reminder**
> >
> > Dear Reviewer,
> >
> > as the rebuttal period ends soon, we hope our responses to your review have addressed your concerns. We’d greatly value your input to ensure the manuscript fully reflects your expectations - please let us know if there are additional points we can clarify or refine.

---

> > > ### Comment · Reviewer_2WJ3 · 2024-11-26
> > >
> > > I have carefully reviewed the authors' answers (both general answers and answers for other reviewers). However, I don't think my concerns have been resolved.
> > > The authors have suggested a good method and good explanations, but unfortunately, I'm sorry to say that I don't think I can change my score.
> > >
> > > Let me explain the points that need to be improved in more detail as follows.
> > >
> > > 1. The biggest part is that the performance improvement is not significant when checked through Tables 1 and 2. It seems difficult to say that there is a clear performance improvement compared to the existing XGB and CatBoost, and so on.
> > >
> > > 2. It seems that the novelty mentioned in the answer is still not well explained. There is a lack of logical explanation about why the sequential method is meaningful, what advantages Mamba was introduced to utilize, and what parts were modified to fit tabular data and on what basis when Mamba was introduced. It seems that in the paper it only explains that the performance was good when applied. A more logical explanation should be provided, and related evidence and experiments should be supported.
> > > 3. The authors' response stated that the advantages of incremental integration of additional features are great, but there is not enough evidence or experiments related to this.
> > >
> > > The presentation also needs to be improved, and when the contents mentioned above are supplemented well, it is expected to be a good paper.

---

> > > > ### Author Response · Authors · 2024-11-27
> > > > **Answer II**
> > > >
> > > > Dear reviewer,
> > > >
> > > > Thank you for your response!
> > > >
> > > > > I have carefully reviewed the authors' answers (both general answers and answers for other reviewers). [...] The authors have suggested a good method and good explanations.
> > > >
> > > > - Thank you for acknowledging our response and its quality.
> > > >
> > > > > The performance improvement is not significant when checked through Tables 1 and 2. It seems difficult to say that there is a clear performance improvement compared to the existing XGB and CatBoost, and so on.
> > > >
> > > > - Thank you for this comment. We understand there is often a focus in reviews—particularly at venues like ICLR—on expanding benchmarks with ever-larger studies, including more models and datasets. However, we would like to emphasize that our method is deliberately unconventional, applying techniques traditionally considered unsuitable for tabular tasks. The fact that it performs comparably to specialized models like XGBoost and CatBoost highlights its potential and demonstrates the broader possibilities for sequential models in this domain.
> > > > We are genuinely surprised and intrigued that Mambular—a fully sequential model—achieves performance comparable to the tested baselines. We hope this sparks interest in exploring such unconventional approaches further.
> > > > Hence, our method does not claim to outperform state-of-the-art models like XGBoost or CatBoost in every scenario, its contribution lies in showcasing how sequential models, traditionally considered unsuitable for tabular data, can achieve competitive performance.
> > > >
> > > >
> > > > > It seems that the novelty mentioned in the answer is still not well explained. There is a lack of logical explanation about why the sequential method is meaningful, what advantages Mamba was introduced to utilize, and what parts were modified to fit tabular data and on what basis when Mamba was introduced. It seems that in the paper it only explains that the performance was good when applied.
> > > > A more logical explanation should be provided, and related evidence and experiments should be supported.
> > > >
> > > > - Respectfully, we disagree with this assessment. To our knowledge, this is the first work to successfully apply a fully recurrent model to tabular tasks. The novelty lies in proving that sequential models, which inherently assume ordered input, can be effectively adapted for data without natural ordering. This is not just a technical adjustment—it is a paradigm shift that challenges the long-held assumption that such models are unsuitable for tabular data.
> > > >
> > > > - Section 2 of the paper details the methodology, specifically highlighting how convolutional layers preprocess features to simulate positional invariance before sequential processing begins. This approach fundamentally reconsiders how sequential information processing can be repurposed for tabular tasks.
> > > > The provided ablation study further analyzes different architectures and supports the idea we present: accounting for positional invariance via simple convolutional layers makes sequential models applicable to data that lacks natural ordering. Appendix B also provides deeper insights into how sequence ordering might affect performance.
> > > >
> > > >
> > > >
> > > > While we cannot provide a purely theoretical explanation for why a sequential model outperforms an MLP or ResNet in certain scenarios, we hypothesize that the advantage lies in the sequential updating of a hidden state.
> > > > Mambular iteratively updates a hidden state $ \mathbf{h}\_j $ through learned dynamics:  $\mathbf{h}\_j = \exp\left(\mathbf{\Delta} \odot\_3 \mathbf{A} \right)\_{:, j, :, :}
> > > > \odot\_{1,2,3} \mathbf{h}\_{j-1} + \left (\left( \mathbf{\Delta} \odot\_{1,2} \mathbf{B} \right) \odot\_{1,2,3} \bar{\mathbf{z}}\right )\_{:, j, :, :}.$ Here, the state transition matrix $ \mathbf{A} $ governs the transformation of the hidden state from the previous time step to the current one, the input-feature matrix $ \mathbf{B} $ maps the input features to the hidden state space, and $ \mathbf{\Delta} $ acts as a gating mechanism, controlling the flow of information. This formulation allows the hidden state to capture both local feature information and global dependencies iteratively. Through this sequential refinement, interactions between features are modeled dynamically, enabling the network to adaptively prioritize critical features and their relationships at each step.
> > > > The structured parameterization of $ \mathbf{A} $, often diagonal, ensures gradient stability across layers. By constraining the recurrence dynamics, Mambular avoids vanishing or exploding gradients during training, allowing stable updates to the hidden state.
> > > > Furthermore, Mambular addresses the lack of natural ordering in tabular data through a simple 1D convolutional layer. Please refer to Section 2, page 5 for more details. This step guarantees that the model’s performance remains unaffected by the arbitrary order of features. Further analysis of this invariance is provided in Appendix B.

---

> > > > > ### Author Response · Authors · 2024-11-27
> > > > > **Answer II Contd.**
> > > > >
> > > > > > The authors' response stated that the advantages of incremental integration of additional features are great, but there is not enough evidence or experiments related to this.
> > > > >
> > > > > - Thank you for this comment.
> > > > > Incremental integration is discussed in the context of practical applicability, offering flexibility for scenarios involving dynamic or evolving feature sets. While further experiments on this aspect could enhance the paper, they are beyond the scope of this manuscript. None of the baseline models support feature incremental learning, and addressing this would require introducing a new methodology, new benchmark models, and a completely new conceptual framework in the introduction. This would significantly exceed the scope and page limits of the current manuscript.
> > > > >
> > > > > > The presentation also needs to be improved, and when the contents mentioned above are supplemented well, it is expected to be a good paper.
> > > > > - Thank you for this comment. Could you be a little more specific with regard to this? I.e. did you notice any mistakes in notation or are you referring to a specific section which needs more refinement?
> > > > >
> > > > > ---
> > > > > We thank you again for your valuable feedback and hope that our answers have addressed your concerns.

---

> > > > > > ### Author Response · Authors · 2024-11-30
> > > > > > **Invitation to continue discussion**
> > > > > >
> > > > > > Dear Reviewer,
> > > > > >
> > > > > > We would like to kindly remind you that the extended rebuttal period will end in less than 72 hours.
> > > > > >
> > > > > > We have done our best to address your remaining concerns and answer your questions. We would greatly appreciate it if you could provide any feedback on remaining concerns.

---

### Official Review · Reviewer_TNBh · 2024-11-08

**Soundness:** 3
**Presentation:** 3
**Contribution:** 3
**Rating:** 3
**Confidence:** 5

**Summary:**

The authors propose Mambular, an adaption of the Mamba architecture for tabular data. The authors compare Mambular against various well-known model families in the tabular domain, in both regression and classification tasks, showcasing the competitive performance of the proposed method. The authors additionally perform a series of ablations to provide insights on several design choices.

**Strengths:**

- The work is well written.
- The work ablates several design choices of the proposed method.
- The proposed method achieves competitive performance compared to the considered baselines.
- The work considers both regression and classification tasks.

**Weaknesses:**

- The related work section misses core references and can be further strengthened. [1][2]
- Line 279, I would rather the authors advocated that the results are not significant, rather than considering a 10% significance level. The standard is a 5% significance level.
- The authors do not perform hyperparameter tuning.
- It is not clear how the set of defaults for all methods is devised.
- The number of datasets considered is limited. Additionally, the authors do not use well-established benchmarks from the community. [3][4]
- A detailed analysis regarding time is not provided to have a clear understanding of the pros and cons of different methods.

[1] Prokhorenkova, L., Gusev, G., Vorobev, A., Dorogush, A. V., & Gulin, A. (2018). CatBoost: unbiased boosting with categorical features. Advances in neural information processing systems, 31.

[2] Kadra, A., Lindauer, M., Hutter, F., & Grabocka, J. (2021). Well-tuned simple nets excel on tabular datasets. Advances in neural information processing systems, 34, 23928-23941.

[3] Gijsbers, P., Bueno, M. L., Coors, S., LeDell, E., Poirier, S., Thomas, J., ... & Vanschoren, J. (2024). Amlb: an automl benchmark. Journal of Machine Learning Research, 25(101), 1-65.

[4] Grinsztajn, L., Oyallon, E., & Varoquaux, G. (2022). Why do tree-based models still outperform deep learning on typical tabular data?. Advances in neural information processing systems, 35, 507-520.

**Questions:**

- Line 47, missing related work [1].
- Line 252, missing related work [2].
- Line 264, how was the set of defaults devised? Do the authors use the defaults of the baselines from the corresponding papers?
- Line 271, how were the datasets selected?
- In terms of baselines, why was CatBoost not included? Given that it handles categorical variables natively? It is additionally among the best performing methods from the gradient-boosted decision tree family.
- Could the authors apply significance tests over the entire suite of datasets and not on a per-dataset basis? The mean value could be used to aggregate the multiple runs on a single dataset. Additionally, could all of the methods be considered and a critical difference diagram be provided?
- Line 297, the results do not align with the referenced work, in particular in [3], (Table 1, Table 2, Figure 3) the FT-Transformer architecture lags behind the ResNet architecture in terms of performance. Additionally, as I previously mentioned, CatBoost is the top performing method, which the authors do not include in the comparisons.
- A comparison with TabPFN for datasets that fit the limitations of the method would be interesting.
- What is the runtime of the proposed method compared to the baselines? What about the inference time?

[1] Kadra, A., Lindauer, M., Hutter, F., & Grabocka, J. (2021). Well-tuned simple nets excel on tabular datasets. Advances in neural information processing systems, 34, 23928-23941.

[2] Prokhorenkova, L., Gusev, G., Vorobev, A., Dorogush, A. V., & Gulin, A. (2018). CatBoost: unbiased boosting with categorical features. Advances in neural information processing systems, 31.

[3] McElfresh, D., Khandagale, S., Valverde, J., Prasad C, V., Ramakrishnan, G., Goldblum, M., & White, C. (2024). When do neural nets outperform boosted trees on tabular data?. Advances in Neural Information Processing Systems, 36.

---

> ### Author Response · Authors · 2024-11-20
> **Response to Questions and Weaknesses**
>
> Dear Reviewer,
>
> Thank you for your thoughtful and constructive feedback. We greatly appreciate the time and effort you have invested in providing valuable insights. Below, we have carefully addressed your comments.
>
> ## Weaknesses
>
> > The related work section misses core references and can be further strengthened. [1, 2]
>
> - Thank you for highlighting this. We will incorporate these references, as well as other relevant literature, such as [3, 4], to ensure comprehensive coverage in the related work section.
>
> > Line 279, I would rather the authors advocated that the results are not significant, rather than considering a 10% significance level. The standard is a 5% significance level.
>
> - We draw your attention to Table 1, where we report significant differences at the standard 5% significance level. The 10% level was included to provide additional context, but our primary interpretation adheres to the standard 5%.
>
> > The authors do not perform hyperparameter tuning.
>
> - Please refer to Section 5, where we discuss our approach in detail. For all baseline methods, we use default parameters established in the literature i.e. [5]. These parameters have been shown to perform well for the baseline models, and tuning would likely only further improve Mambular's performance relative to the baselines, as these baselines have had years of optimization while Mambular has not yet received the same level of fine-tuning.
>
> > It is not clear how the set of defaults for all methods is devised.
>
> - We utilize the default settings as defined in the original publications for each baseline method. For additional clarity, see [5] and Section E in the appendix, as well as the config files provided in our codebase, where we clearly specify each parameter.
>
> > The number of datasets considered is limited. Additionally, the authors do not use well-established benchmarks from the community.
>
> - We address this point in our general response. Additionally it is worth noting for the suggested [5] that all 176 datasets are classification tasks, with nearly half containing fewer than 2000 observations. Training deep learning models with significantly more parameters than data points seems a bit unreasonable.
>
> > A detailed analysis regarding time is not provided to have a clear understanding of the pros and cons of different methods.
>
> - Thank you for the suggestion. We agree that runtime analysis is valuable, especially when comparing Mambular with transformer-based architectures. The bottlenecks for both are well-known—linear increase in memory consumption for Mambular and quadratic for transformers. Notably, Mambular outperforms FTTransformer with the same number of parameters when the number of features is ≥35 on an Nvidia T4 GPU.
>
>
>
> ## Questions
>
> > In terms of baselines, why was CatBoost not included?
>
> - Thank you for bringing that to our attention. We initially excluded CatBoost, as XGBoost and LightGBM outperformed it in prior work [3]. However, we understand the value of broader comparisons and have included CatBoost as suggested.
>
> > Could the authors apply significance tests over the entire suite of datasets and not on a per-dataset basis?
>
> - Thank you for this comment. As noted in our general response, we find that Mambular does not significantly underperform or outperform CatBoost across the suite of datasets. However, we would like to reemphasize that the primary focus of our study is to demonstrate the viability of truly sequential models for tabular tasks—an idea that has been largely overlooked in prior research.
>
> > Line 297, the results do not align with the referenced work, in particular in [6].
>
> - The benchmarks presented in [6] do not align with our findings, nor with results from other independent benchmarks such as [3, 4, 5]. We address our reasoning in greater detail in our general response to provide a critical analysis of the results in [6].
>
> > A comparison with TabPFN for datasets that fit the limitations of the method would be interesting.
>
> - While we appreciate the suggestion, none of our datasets satisfy TabPFN's constraints. Additionally, TabPFN is limited to classification tasks, making it unsuitable for the full scope of our experiments.
>
> ---
>
> We hope these clarifications effectively address your concerns and provide additional insights into our methodologies and choices. Thank you again for your feedback.
>
> ---
> [1] Prokhorenkova, Liudmila, et al. "CatBoost: unbiased boosting with categorical features."
> [2] Kadra, Arlind, et al. "Well-tuned simple nets excel on tabular datasets."
> [3] Rubachev, Ivan, et al. "TabReD: Analyzing Pitfalls and Filling the Gaps in Tabular Deep Learning Benchmarks."
> [4] Gorishniy, Yury, et al. "TabM: Advancing Tabular Deep Learning with Parameter-Efficient Ensembling."
> [5] Grinsztajn, Leo, Eet al. "Why do tree-based models still outperform deep learning on typical tabular data?."
> [6] McElfresh, Duncan, et al. "When do neural nets outperform boosted trees on tabular data?."

---

> > ### Author Response · Authors · 2024-11-24
> > **Rebuttal Reminder**
> >
> > Dear Reviewer,
> >
> > as the rebuttal period ends soon, we hope our responses to your review have addressed your concerns. We’d greatly value your input to ensure the manuscript fully reflects your expectations - please let us know if there are additional points we can clarify or refine.

---

> ### Comment · Reviewer_TNBh · 2024-11-27
> **Author Response**
>
> I would like to thank the authors for their reply. I have read the other reviews and their corresponding answers. Below are my comments:
>
> - **Line 263-266: "All neural models share several parameters: starting learning rate of 1e-04, weight decay of 1e-06,
>  a nearly stopping patience of 15 epochs with respect to the validation loss, a maximum of 200 epochs
>  for training, and learning rate decay with a factor of 0.1 with a patience of 10 epochs with respect
>  to the validation loss."**
>
>
>    Following the above, the learning rate, weight decay, training epochs are all hyperparameters that are set to very specific values for all methods. These do not correspond to default hyperparameters suggested on a per-method basis.
>
> - To my understanding the manuscript was revised but the proposed references were not incorporated.
> - Judging by the revised manuscript (Table 2) CatBoost outperforms Mambular.
> - I do not agree with the authors that training deep learning methods with a significant number of parameters compared to data instances is unreasonable, there exist tasks where deep learning methods outperform traditional methods like GBDTs on really small datasets (for example take the case of TabPFN) and tree-based methods outperforming deep learning methods on large datasets.
> - A more extensive number of datasets should be included in the experiments by incorporating the already used benchmarks. Although the authors did provide results, the results are not extensive.
> - Results with hyperparameter tuning should be provided, to show the full potential of the considered methods.
> - Lastly, the authors should provide detailed statistics about the training/inference time of all the considered baselines. As for example, CatBoost outperforms Mambular, if Mambular has a worse training and inference time compared to CatBoost, one might ask why a practitioner would use Mambular instead of CatBoost?
>
> Based on the above, the majority of my points remain unresolved. Based on which I cannot recommend the work for acceptance and I will keep my original score.

---

> ### Author Response · Authors · 2024-11-27
> **Answer II**
>
> Dear Reviewer,
>
> Thank you for your thoughtful comments. We would like to clarify that the primary aim of our manuscript is not to serve as a benchmarking study on tabular deep learning. We understand there may be some misunderstanding regarding the scope and focus of our work, and we kindly encourage you to review both the manuscript and our previous responses for further clarification.
>
> We’ve addressed each point in detail below, including those that may not align directly with the primary focus of our study, and we hope this helps to provide additional clarity.
>
> > Line 263-266: "All neural models share several parameters: starting learning rate of 1e-04, weight decay of 1e-06, an early stopping patience of 15 epochs with respect to the validation loss, a maximum of 200 epochs for training, and learning rate decay with a factor of 0.1 with a patience of 10 epochs with respect to the validation loss." Following the above, the learning rate, weight decay, training epochs are all hyperparameters that are set to very specific valuesfor all methods. These do not correspond to default hyperparameters suggested on a per-method basis.
>
> - Thank you for this comment. We would like to clarify that all hyperparameters, including shared parameters, are clearly described in the manuscript and presented in detail. Additionally, we have provided all configuration files in our codebase to ensure full transparency.  Early stopping is performed based on validation loss, and the best model is selected accordingly. Consequently, the maximum number of epochs has no impact on the results, as no model, on any dataset or fold, reached this limit. The parameters we use, such as learning rate and weight decay, are consistent with those reported in related works (e.g., [1, 2, 3]). This approach was chosen to ensure fairness, reproducibility, and alignment with established practices.
>
> - All architectual hyperparameters are chosen based on well established default values, e.g. for CatBoost, XGBoost, LightGBM or FT-Transformer. Additionally, we would like to reference you to our limitations section were we explicitly regard this.
>
> > To my understanding the manuscript was revised but the proposed references were not incorporated.
>
> - Thank you for this comment. Of the five references you initially suggested, three ([4, 5, 6]) have been incorporated into the revised manuscript. The remaining references were not included as we found them less directly relevant to the scope of our work. However, we are happy to include them in the final version if you think they provide additional value.
>
>
> > Judging by the revised manuscript (Table 2) CatBoost outperforms Mambular.
>
> - Thank you for this observation with which we agree. As reported in Table 2, CatBoost does outperform Mambular in certain cases, which is consistent with the goals and scope of our work. Our manuscript does not claim universal superiority over specialized models like CatBoost or XGBoost.
>
> - Instead, the primary contribution of this work is to demonstrate that sequential models, traditionally deemed unsuitable for tabular data, can achieve competitive performance. The fact that Mambular, a fully sequential model, performs comparably to highly optimized models like CatBoost highlights its potential and broadens the possibilities for applying unconventional approaches to tabular data modeling.
>
> - This work is not intended to replace established methods but rather to expand the toolkit for tabular deep learning, showcasing that unconventional techniques can perform remarkably well. We hope it inspires further exploration of sequential and state-space methods in this context.
>
> > I do not agree with the authors that training deep learning methods with a significant number of parameters compared to datainstances is unreasonable, there exist tasks where deep learning methods outperform traditional methods like GBDTs on really smalldatasets (for example take the case of TabPFN) and tree-based methods outperforming deep learning methods on large datasets.
>
> - Thank you for your comment. However, this point might stem from a misunderstanding and appears to us somewhat out of context from our privious response. To clarify, we referred to one of the benchmarks you suggested, noting as an additional point that this benchmark is limited to classification tasks and, moreover, includes extremely small datasets with fewer than 500 observations. This critique was specific to the benchmark in question and does not pertain to the scope of our manuscript. We would also like to kindly refer you to our general response, where we address the benchmarking concerns in more detail.
>
> - Furthermore, TabPFN operates in a fundamentally different paradigm than general tabular deep learning models, as it is a meta-learner designed for in-context tabular classification. This implies that TabPFn operates under fundamentally different assumptoins making it unsuitable for the tasks addressed by our method.

---

> ### Author Response · Authors · 2024-11-27
> **Answer II Contd.**
>
> > A more extensive number of datasets should be included in the experiments by incorporating the already used benchmarks. Althoughthe authors did provide results, the results are not extensive. Results with hyperparameter tuning should be provided, to show the full potential of the considered methods.
>
> - Please note that we have extensively addressed these concerns in our general response. The additional results will be incorporated into our final version; however, we cannot currently alter this version as this comment was received after the end of the original rebuttal period.
>
> > Lastly, the authors should provide detailed statistics about the training/inference time of all the considered baselines. As forexample, CatBoost outperforms Mambular, if Mambular has a worse training and inference time compared to CatBoost, one might ask whya practitioner would use Mambular instead of CatBoost?
>
> - Thank you again for your comment.  We originally addressed your question in relation to FTTransformer and Mambular only, as we had misunderstood it. Consequently, we did not include details over all models in the revised version and only answered your question directly.
> However, it is well-known that most deep learning methods are slower than GBDTs. This is a common limitation of deep learning approaches, with the exception of specialized architectures like TabPFN, which is not suitable for the benchmarks presented in our work. The strengths of Mambular lie in its ability to handle very large datasets, it scales to a high number of features, and uniquely supports feature incremental learning—capabilities not available in CatBoost or other deep learning methods.
> We will include detailed statistics on training and inference times for all models in the final version of the manuscript.
>
> ---
> We appreciate your comments and understand that your primary concerns focus on the limited benchmarks and the performance relative to other models. While we fully acknowledge the importance of benchmark evaluations, we kindly ask that you refer to our explanation in the general response and consider the wider contributions of our work.
>
> The aim of this manuscript is not simply to create a model that outperforms every existing method on the largest possible benchmarks, but to introduce a novel approach that challenges conventional assumptions about what is possible for tabular data. Our work explores the potential of sequential models - methods typically seen as unsuitable for such tasks - and shows that they can, in fact, achieve competitive performance. This is a significant conceptual advancement, one that opens up new possibilities for applying sequential models in areas where they have traditionally been overlooked.
>
> We hope this perspective encourages you to look beyond the performance metrics and appreciate the theoretical innovation and potential applications our method introduces.
>
> ---
> [1] Gorishniy, Yury, et al. "Revisiting deep learning models for tabular data."
> [2] Popov, Sergei, et al. "Neural oblivious decision ensembles for deep learning on tabular data."
> [3] Gorishniy, Yury, et al. "On embeddings for numerical features in tabular deep learning." NeurIPS (2022)
> [4] McElfresh, Duncan, et al. "When do neural nets outperform boosted trees on tabular data?."
> [5] Grinsztajn, Leo, Eet al. "Why do tree-based models still outperform deep learning on typical tabular data?."
> [6] Prokhorenkova, Liudmila, et al. "CatBoost: unbiased boosting with categorical features."

---

> > ### Author Response · Authors · 2024-11-30
> > **Invitation to continue discussion**
> >
> > Dear Reviewer,
> >
> > We would like to kindly remind you that the extended rebuttal period will end in less than 72 hours.
> >
> > We have done our best to address your remaining concerns and answer your questions. We also kindly draw your attention to our general response, which addresses many of your initial concerns and questions.

---

> > > ### Comment · Reviewer_TNBh · 2024-12-02
> > > **Author Response**
> > >
> > > I would like to thank the authors for the additional reply. However, my concerns have not been addressed.
> > >
> > > > Please note that we have extensively addressed these concerns in our general response. The additional results will be incorporated into our final version; however, we cannot currently alter this version as this comment was received after the end of the original rebuttal period.
> > >
> > > I would like to remind the authors that I have not raised any additional points during the end of the rebuttal period, all of my concerns were raised in my initial review. Be it additional results or hyperparameter optimisation. On my previous answer I merely reiterated on my previous points.
> > >
> > > > Thank you for this comment. Of the six references you initially suggested, three ([4, 5, 6]) have been incorporated into the revised manuscript. The remaining references were not included as we found them less directly relevant to the scope of our work. However, we are happy to include them in the final version if you think they provide additional value.
> > >
> > > In my initial review, I suggested strengthening the related work section, and the authors agreed. If I believed the references lacked additional value, I would not have included them. If the authors disagreed with these suggestions, they could have explained why the references might not be appropriate. Ultimately, this is meant to be a constructive discussion.
> > >
> > >
> > > While I find the work interesting, and agree with the authors that this is not a benchmarking paper, I believe the experimental protocol should be more extensive as I mentioned in my original review. This would give more credibility to the provided results, additionally a thorough analysis should be provided regarding the pros and cons of individual methods. Based on the above I cannot recommend the current version for acceptance.

---

> > > > ### Author Response · Authors · 2024-12-02
> > > > **Answer to Reviewer Response III**
> > > >
> > > > Dear Reviewer,
> > > >
> > > > Thank you for your detailed feedback. We greatly appreciate the time and effort you have dedicated to reviewing our work and would like to address a few points where there may have been some misunderstandings.
> > > >
> > > > > *I would like to remind the authors that I have not raised any additional points during the end of the rebuttal period; all of my concerns were raised in my initial review.*
> > > >
> > > > This appears to stem from a misunderstanding on our part. Specifically, your earlier comment suggested that *"a more extensive number of datasets should be included in the experiments by incorporating the already used benchmarks."* We wanted to clarify that while we fully intend to integrate these results into future iterations of our work, we are unable to submit a revised version at this moment due to the constraints of the review process. Our intention was not to imply that you suggested adding or altering experiments at this stage, and we apologize if our previous response gave a different impression.
> > > >
> > > > > *In my initial review, I suggested strengthening the related work section, and the authors agreed. If I believed the references lacked additional value, I would not have included them. If the authors disagreed with these suggestions, they could have explained why the references might not be appropriate. Ultimately, this is meant to be a constructive discussion.*
> > > >
> > > > As noted, we have integrated the majority of your suggestions into the manuscript. However, regarding the references:
> > > > 1. We found [1] unsuitable for inclusion, as this is not intended to be a benchmarking study. Specifically, a benchmark on AutoML is not relevant to the scope of our work.
> > > > 2. We initially found [2] to be similar to [3] and, therefore, did not consider it meaningful to include. However, upon further inspection, we now recognize that [2] offers an interesting perspective and provides additional insights that complement [3]. As such, we have decided to include it in our revised manuscript.
> > > >
> > > > > *While I find the work interesting, and agree with the authors that this is not a benchmarking paper, I believe the experimental protocol should be more extensive, as I mentioned in my original review. This would give more credibility to the provided results. Additionally, a thorough analysis should be provided regarding the pros and cons of individual methods.*
> > > >
> > > > Firstly, we thank you for recognizing the merit of our work and appreciate your acknowledgment that this is not a benchmarking study. While we have provided detailed analyses of the advantages and disadvantages of our method in Sections 4, 5, and 6, we reiterate that extending these analyses to cover all other methods would shift the focus toward a benchmarking paper, which is not the intent of our work.
> > > >
> > > > Furthermore, the ablation study highlights the dependence on convolutional dimensions and addresses issues related to sequence ordering, pooling, cls tokens, bidirectional processing and the inclusion of learnable feature interaction layers. Section 6 (Limitations) directly addresses concerns about hyperparameter optimization, and the benefits stemming from Mamba's efficiency are well-documented in the literature, rendering additional reiteration in our manuscript unnecessary [4, 5].
> > > >
> > > > We have carefully addressed these concerns in our general response and throughout the manuscript. While we acknowledge the broader value of benchmarking in ML research, we find this expectation can sometimes detract from exploring deeper methodological contributions. We respectfully ask that our work be evaluated based on its intended scope and contributions, as outlined in our response.
> > > >
> > > > Thank you again for your time and valuable feedback.
> > > >
> > > > ---
> > > > [1] Gijsbers, P., Bueno, M. L., Coors, S., LeDell, E., Poirier, S., Thomas, J., ... & Vanschoren, J. (2024). Amlb: an automl benchmark. Journal of Machine Learning Research, 25(101), 1-65.
> > > > [2] Kadra, A., Lindauer, M., Hutter, F., & Grabocka, J. (2021). Well-tuned simple nets excel on tabular datasets. Advances in neural information processing systems, 34, 23928-23941.
> > > > [3]  Gorishniy, Yury, et al. "Revisiting deep learning models for tabular data." NeurIPS (2021).
> > > > [4] Gu, Albert, and Tri Dao. "Mamba: Linear-time sequence modeling with selective state spaces." arXiv preprint arXiv:2312.00752 (2023).
> > > > [5] Dao, Tri, and Albert Gu. "Transformers are SSMs: Generalized models and efficient algorithms through structured state space duality." ICML (2024).
> > > >
> > > > ---

---

### Author Response · Authors · 2024-11-20
**General Response**

Dear Reviewers,

We would like to extend our sincere gratitude for your careful consideration of our work. After reviewing all feedback, we observed several consistent points that we would like to address in a joint response. Overall, we were somewhat surprised by the low scores, especially since the primary critique from most reviewers focuses on a single aspects of our contribution.

## Benchmarks
Nearly all reviewers have suggested expanding the benchmarks, citing studies such as [1, 2, 3]. While additional experiments would indeed strengthen any study, we respectfully urge you to consider the following points:

- The referenced studies [1, 2, 3] are large-scale benchmark studies, some involving over 150 datasets. These benchmarks require extensive hyperparameter optimization and k-fold cross-validation. Given our computational resources, reproducing such an extensive benchmark is infeasible, particularly for deep learning (DL) models on tabular data, which demand considerable computational power. In the case of [2], for example, even a WandB subscription is required. This presents a barrier in research publication, where access to substantial compute resources has become an unspoken prerequisite. We find that the pressure for "more experiments" has become a generic critique in ML research, while core aspects such as reproducibility often go overlooked. While we cannot feasibly match the requested benchmark scale, our codebase is comprehensive, and we provide detailed configuration files for all models to ensure full reproducibility.

- To still provide a robust experimental evaluation, we conducted a benchmark using datasets sourced from [OpenML](https://www.openml.org/search?type=study&study_type=task&sort=tasks_included&id=353) and presented in [4], with missing values removed and datasets with fewer than 1000 observations omitted. Currently, we have run the best models from our manuscript as well as the suggested Catboost [6] and achieved the results detailed below. Catboost does indeed outperform XGboost, contrary to our expectations given results from [3] and [5]. Catboost performs on average, over all datasets practially identical to Mambular.





| Model             |      BH ↓  |    CW ↓   |   FF ↓   |     GS ↓  |       HI ↓ |   K8 ↓   |     AV ↓  |      KC ↓  |     MH ↓  |       NP ↓ |      PP ↓ |   SA ↓   |   SG ↓     |       VT ↓| Rank ↓    |
|:------------------|-----------:|----------:|---------:|----------:|-----------:|---------:|----------:|-----------:|----------:|-----------:|----------:|---------:|-----------:|----------:|----------:|
| Mambular          |  **0.021** | **0.701** |  0.272   |     0.057 |  **0.595** |    0.168 |     0.018 |      0.137 |     0.085 |  **0.003** |     0.402 |**0.015** |      0.318 | **0.003** | **1.79**  |
| FTTransformer     |      0.028 |     0.701 |  0.301   |     0.205 |      0.609 |    0.451 |     0.089 |      0.149 |     0.101 |      0.009 |     0.542 |    0.033 |      0.36  |     0.045 |     4.36  |
| CatBoost          |      0.032 |     0.702 |**0.245** | **0.041** |      0.597 |**0.150** |     0.004 |  **0.110** | **0.078** |      0.005 | **0.39**  |    0.018 |  **0.297** |     0.013 | **1.79**  |
| LightGBM          |      0.048 |     0.707 |  0.263   |     0.059 |      0.599 |    0.239 |     0.024 |      0.140 |     0.091 |      0.009 |     0.452 |    0.031 |      0.302 |     0.013 |     3.26  |
| XGBoost           |      0.039 |     0.752 |  0.281   |     0.078 |      0.635 |    0.259 | **0.004** |      0.161 |     0.098 |      0.006 |     0.403 |    0.024 |      0.329 |     0.013 |     3.71  |


We plan to incorporate additional models and benchmarks gradually. However, given our computational constraints, we performed a single train-test-validation split for this evaluation. All necessary files to reproduce these benchmarks are available in our codebase.

---

> ### Author Response · Authors · 2024-11-20
> **General Response Contd.**
>
> Additionally, we fitted NODE [5] and CatBoost [6] on the current benchmark of the paper, with results shown below:
>
> | Models    | DI ↓              | AB ↓              | CA ↓              | WI ↓             | PA ↓             | HS ↓             | CP ↓             | BA ↑             | AD ↑             | CH ↑             | FI ↑             | MA ↑              | Rank ↓     |
> |-----------|-------------------|-------------------|-------------------|------------------|------------------|------------------|------------------|------------------|------------------|------------------|------------------|-------------------|------------|
> | Mambular  | **0.018** ± 0.000 | 0.452     ± 0.043 | **0.167** ± 0.011 | 0.628 ± 0.010    | **0.035** ± 0.005| 0.132 ± 0.020    | 0.025 ± 0.002    | 0.927 ± 0.006    | **0.928** ± 0.002| 0.861 ± 0.008    | **0.796** ± 0.013| 0.917  ± 0.003    |  **1.75**  |
> | NODE      | 0.019     ± 0.000 | **0.431** ± 0.052 | 0.207     ± 0.001 | 0.613 ±     0.006| 0.045 ± 0.007    | 0.124     ± 0.015| 0.026 ± 0.001    | 0.914 ± 0.008    | 0.904 ± 0.002    | 0.851 ± 0.006    | 0.790 ± 0.010    | 0.904  ± 0.005    |  2.5       |
> | CatBoost  | 0.019     ± 0.000 | 0.457 ± 0.054     | 0.169 ± 0.007     | **0.583** ± 0.006| 0.045 ± 0.006    | **0.106** ± 0.015| **0.022** ± 0.001| **0.932** ± 0.008| 0.927 ± 0.002    | **0.867** ± 0.006| 0.796 ± 0.010    | **0.926**  ± 0.005|  **1.75**  |
>
>
>
>
>
> ## Novelty
> We understand that several reviewers have raised concerns about the novelty of our approach. We would like to clarify the following points:
>
> - To the best of our knowledge, our model is the first model to explicitly interpret tabular data as a sequence, enabling the application of a wide range of sequential methods to tabular data. This approach is particularly novel because, while tabular data typically lacks an inherent ordering, sequential models still demonstrate strong performance. This insight opens up new possibilities, particularly in feature-incremental learning.
>
> - For practitioners working with large datasets, Mambular offers a distinct advantage by allowing incremental integration of additional features without requiring complete retraining. In contrast, the other methods we compared necessitate full retraining with each update.
>
> Thank you once again for your valuable feedback. We hope our responses address your concerns and underscore the significance of our contributions.
>
>
>
> ---
> [1] McElfresh, Duncan, et al. "When do neural nets outperform boosted trees on tabular data?."
> [2] Grinsztajn, Leo, Eet al. "Why do tree-based models still outperform deep learning on typical tabular data?."
> [3] Rubachev, Ivan, et al. "TabReD: Analyzing Pitfalls and Filling the Gaps in Tabular Deep Learning Benchmarks."
> [4] Fischer, Sebastian, et al. "A curated tabular regression benchmarking suite."
> [5] Popov, Sergei, et al. "Neural oblivious decision ensembles for deep learning on tabular data."
> [6] Prokhorenkova, Liudmila, et al. "CatBoost: unbiased boosting with categorical features."

---

### Meta-Review · Area_Chair_77Kv · 2024-12-19

**Metareview:**

The authors apply the Mamba architecture to tabular data. The proposed technique uses memory more efficiently than transformer-based models and performs about as well as the selected baselines.

However, reviewers expressed concerns that evaluation is limited. In addition, no thorough hyperparameter tuning is performed, and the number of datasets and baselines is small. The authors do not exhaustively explain why a sequential model would be a good fit for tabular data in the first place.

Overall the work lacks novelty, the analysis is not sufficiently detailed, and the experimental protocol is sub-optimal (in terms of datasets, and baselines). Although the authors added a few more results in their rebuttal, these new findings did not change the essence of the quality of the manuscript.

I suggest rejecting this submission for now and encouraging the authors to incorporate the reviewers’ feedback into a future version.

**Additional Comments On Reviewer Discussion:**

The reviewers engaged with the authors during the rebuttal, however, some of the reviewers' questions remained not thoroughly answered.

---

### Decision · Program_Chairs · 2025-01-22

Reject